

Characterization of lower-cost medium precision atmospheric $CO_2$ monitoring systems for urban
areas using commercial NDIR sensors
Emmanuel Arzoumanian[1], Felix R. Vogel[1,2*], Ana Bastos[1], Bakhram Gaynullin[3], Olivier Laurent[1],
Michel Ramonet[1] and Philippe Ciais[1*]
1 LSCE/IPSL, CEA-CNRS-UVSQ, Universite Paris-Saclay, Gif-Sur-Yvette, France
2 Climate Research Division, Environment and Climate Change Canada, Toronto, Canada
3 SenseAir AB, Delsbo, Sweden
* Corresponding authors: Felix.Vogel@canada.ca and Philippe.Ciais@lsce.ipsl.fr
**Abstract:**
$CO_2$ emission estimates from urban areas can be obtained with a network of in-situ instruments
measuring atmospheric $CO_2$ combined with high-resolution (inverse) transport modeling. The
distribution of $CO_2$ emissions being highly heterogeneous in space and variable in time in urban
areas, gradients of atmospheric $CO_2$ need to be measured by numerous instruments placed at
multiple locations around and possibly within these urban areas, which calls for the development
of lower-cost medium precision sensors to allow a deployment at required densities. Medium
precision is here set to be a random error (uncertainty) on hourly measurements of ±1 ppm or
less, a precision requirement based on previous studies of network design in urban areas. Here
we present tests of a HPP commercial NDIR sensors manufactured by Senseair AB performed
in the laboratory and at actual field stations, the latter for $CO_2$ concentration in the Paris area.
The lower-cost medium precision sensors are shown to be sensitive to atmospheric pressure
and temperature conditions. The sensors respond linearly to $CO_2$ when measuring calibration
tanks, but the regression slope between measured and true $CO_2$ differs between individual
sensors and changes with time. In addition to pressure and temperature variations, humidity
impacts the measurement of $CO_2$, all causing systematic errors. In the field, an empirical
calibration strategy is proposed based on parallel measurements with the lower-cost medium
precision sensors and a high-precision instrument cavity ring-down instrument during 6 month.
This empirical calibration method consists of using a multiple regression approach to create a
model of the errors defined as the difference of $CO_2$ measured by the lower-cost medium
precision sensors relative to a calibrated high-precision instrument, based on predictors of air
temperature, pressure and humidity. This error model shows good performances to explain the
observed drifts of the lower-cost medium precision sensors on time scales of up to 1-2months
when trained against 1-2 weeks of high-precision instrument time series. Residual errors are
contained within the ±1 ppm target, showing the feasibility to use networks of HPP instruments
for urban $CO_2$ networks, provided that they could be regularly calibrated against one anchor
reference high-precision instrument.

**1. Introduction**
Urban areas cover only a small portion (< 3 %) of the land surface but account for about 70% of
fossil fuel $CO_2$ emissions (Liu et al. 2014, Seto et al. 2014). Uncertainties of fossil fuel $CO_2$
emissions from inventories based on statistics of fuel amounts and/or energy consumption are
on the order of 5% for OECD countries and up to 20% in other countries (Andres et al. 2014) but
they are larger in the case of cities (Breon et al. 2015, Wu et al. 2016). Further, in many cities of
the world, there are no emission inventories available. The need to provide more reliable



information on emission and emission trends has prompted research projects seeking to provide
estimates of GHG budget cities, power plants and industrial sites, based on in situ
measurements made at surface stations (Staufer et al. 2016, Lauvaux et al. 2016, Verhulst al.
2017), aircraft campaigns around emitting locations Mays et al. 2009, Cambaliza et al. 2014)
and satellite imagery (Broquet et al. 2017, Nassar et al. 2017). Although sampling strategies and
measurement accuracies differ between these approaches, the commonly used principle is to
measure atmospheric $CO_2$ mixing ratio gradients at stations between the upwind and downwind
vicinity of an emitting area and infer the emissions that are consistent with those $CO_2$ gradients
and their uncertainties, using an atmospheric transport model. This approach is known as
atmospheric $CO_2$ inversion or as a "top-down" estimate.
Inversion studies from Paris, France attempting to constrain $CO_2$ emissions from measurements
of $CO_2$ mixing ratios at four stations located around the city along the dominant wind direction,
have pointed out that the fast mixing by the atmosphere and the complex structure of urban $CO_2$
emissions requires high resolution atmospheric transport models, and continuous
measurements of the atmosphere to select gradients induced by emission plumes (Broquet et al.
2015, Wu et al. 2016) that can be captured at the scale of the model.
With four stations, only the $CO_2$ emissions from the Paris megacity could be retrieved with an
accuracy of ≈20% on monthly budgets (Wu et al. 2016). A denser network of stations would help
to obtain more information on the spatial details of $CO_2$ emissions. A network design study by
Wu et al. 2016 for the retrieval of $CO_2$ emissions per sector for the Paris Megacity has shown
that with 10 stations measuring $CO_2$ with 1 ppm accuracy on hourly time-steps, the error of the
annual emission budget could be reduced down to a 10% uncertainty. Wu et al. 2016
furthermore found that for a more detailed separation of emissions into different sectors, more
stations were needed, on the order of 70 stations to be able to separate road transport from
residential $CO_2$ emissions. This inversion based on pseudo-data allowed estimating total $CO_2$
emissions with a better accuracy than 10% and emissions of most major source sectors
(building, road energy) with an accuracy better than 20%. Another urban network design study
over the San Francisco Bay area reached a similar conclusion, i.e. that in-situ $CO_2$
measurements from 34 stations with 1 ppm accuracy at an hourly resolution could estimate
weekly $CO_2$ emissions from the city area with less than 5 % error (Turner et al. 2016).
In the studies from Wu et al. 2016 and Turner et al. 2016, the additional number of atmospheric
$CO_2$ measurement stations rather than the individual accuracy of each measurement helped to
constrain emissions, provided that $CO_2$ observation errors have random errors of less than 1
ppm on hourly measurements, uncorrelated in time, and in space between stations. Therefore,
we will adopt here a 1 ppm uncertainty on hourly $CO_2$ data as the target performance for new
urban lower-cost medium precision CO2 sensors.
Today, the continuous $CO_2$ gas analyzers used for continental scale observing systems like
ICOS (https://www.icos-ri.eu/), NOAA (https://www.esrl.noaa.gov/gmd/) or ECCC
(https://www.canada.ca/en/environment-climate-change.html) follow the WMO/GAW guidelines
and are at least ten times more precise than our target of 1 ppm, but are also quite expensive.
Because for urban inversions, the number of instruments is more important than their individual
precision, if low-cost sensors could be produced with the specifications of 1 ppm random error,
significant expansion of urban networks could be achieved at an acceptable cost.
Recently, inexpensive sensors, measuring trace gases, particulate matter, as well as traditional
meteorological variables, using various technologies and accuracy have become commercially
available. Evaluation and implementation of these sensors is quite promising (Eugster and Kling
2012, Holstius et al. 2014, Piedrahita et al. 2014, Young et al. 2014, Wang et al. 2015, Martin et
al. 2017). With the advent of low cost mid-IR light sources and detectors, different non-
dispersive infrared (NDIR) $CO_2$ sensors have become commercially available.
In this study, we present the development and stability tests of an inexpensive instrument to
measure $CO_2$ based on controlling parameters for ambient air using a Senseair HPP NDIR



sensor for $CO_2$ measurements (Hummelgard et al. 2015). The instrument sensitivities to ambient air temperature, pressure and water vapor content are assessed in laboratory experiments and climate chambers tests. Then, the instrument linearity is evaluated against a suite of $CO_2$ reference gases calibrated from 330 to 1000 ppm. The calibrated low-cost medium precision (LCMP) instruments are then compared to highly precise CRDS instruments (G2401, Picarro Inc, Santa Clara, USA).

Lastly we present the time series of ambient air $CO_2$ measurements in the Paris region environment. The time series are compared to co-located CRDS-based $CO_2$ observations, and Empirical corrections to the HPP-based instrument are proposed based on $CO_2$ and meteorological variables. These corrections established during a period of 1 or 2 weeks are used to estimate the drift of the HPP-instrument on time scales of up to a month and a half.

## 2. Sensor integration

### 2.1. HPP sensor

The HPP (High-Performance Platform) NDIR $CO_2$ sensor from SenseAir AB (Delsbo, Sweden) is a commercially available lower-cost system (Hummelgard et al. 2015). The main components of this sensor are: an infrared source (lamp), a sample chamber (ca. 1m optical path length), a light filter and an infrared detector. The gas in the sample chamber causes absorption of specific wavelengths (Hummelgard et al. 2015) according to the Beer–Lambert law, and the attenuation of light in these wavelengths is measured by a detector to determine the gas mixing ratio. The detector has an optical filter in front of it that eliminates all light except the wavelength that the selected gas molecules can absorb. The HPP has a factory pre-calibrated $CO_2$ measurement range of 0 to 1000 ppm. The HPP sensor itself uses ca. 0.6 W and requires 12 VDC and has a life expectancy superior to 15 years according to the manufacturer.

Three generation of HPP sensors were built by SenseAir AB (Delsbo, Sweden), in this manuscript we only report on the tests carried out on the latest generation (HPP3) being the most performant among the HPP sensors family. Previous HPP versions were used for more short-term airborne measurements, for example in the COCAP system (Kunz et al. 2017) and were found to have an accuracy of 1-1.2 ppm during short-term mobile campaigns.

A number of technical improvements have been made for the new (third) HPP generation described here:

- Simple interface through USB connection and the development of a new software made data transfer easier, quicker and more efficient
- Improved temperature stability due to 6 independent heaters dispatched inside the unit.
- The sample cell design is optimized in order to match with targeted production costs. The optical length was slightly reduced to 95cm so the related alcohol sensor platform could be used to benefit from any development of this product line.
- To improve long-term drift the sensor is equipped with new electronics and the IR sources were preconditioned prior to shipment.
- The improved second version of HPP3 (HPP3.2) sensors was equipped with a pressure sensor (LPS331AP - ST Microelectronics, Switzerland) to allow real-time corrections.
- Leakage problems impact are minimized since the third generation sensor works in a high pressure mode. A pump is thus needed upstream of the sensor inlet in order to create a high pressure in the measurement cell.





Different sensors from two versions of HPP3 were tested and used in this study, that is, three
sensors from a first version (HPP3.1) named S1.1, S1.2 and S1.3, and three others from the
second better version (HPP3.2) named S2.1, S2.2 and S2.3.
**2.2. Portable integrated instrument**
The HPP sensors were integrated into a custom-built portable instrument suitable to perform in-
situ $CO_2$ measurements on ambient air. The instrument is composed of the HPP $CO_2$ sensor,
temperature (T) and relative humidity (RH) sensors. For pressure, a LPS331AP (ST
Microelectronics, Switzerland) sensor was added inside the HPP3.2 sensors. LPS331AP has a
pressure range of 0.26 to 1.26 atm, a high resolution mode of $2\times10^{-5}$ atm Root Mean Square
(RMS) and low power consumption (30 μA for high resolution mode). For humidity and
temperature, a DHT22 sensor (Adafruit, USA) was added in the instrument enclosure and
connected through an I2C interfaces. The accuracy of the sensor is ±2% RH and ±0.5°C. Its
range is 0-100 %RH and -40 to +80°C, respectively.
A Raspberry Pi3 (RPi3) (Raspberry Pi Foundation, 2015) is used to collect the data of all
sensors. The RPi3 is a small (85x56 mm) processor running with Rasbian OS which is a Linux
distribution. It has 40-pin extended GPIO which allows connecting number of sensors, 4 USB2
ports, BCM43438 wireless LAN and Bluetooth low energy on board.
A 7'' touch screen monitor is connected via an adapter board which handles power and signal
conversion. To be able to flush the measurement cell a diaphragm micro pump
(GardnerDenverThomas, USA, Model 1410VD/1.5/E/BLDC/12V) with an external speed adaptor
is used. A 12 V power supply is sufficient to power the integrated package. An image of the
complete portable instrument package is available in Figure1.

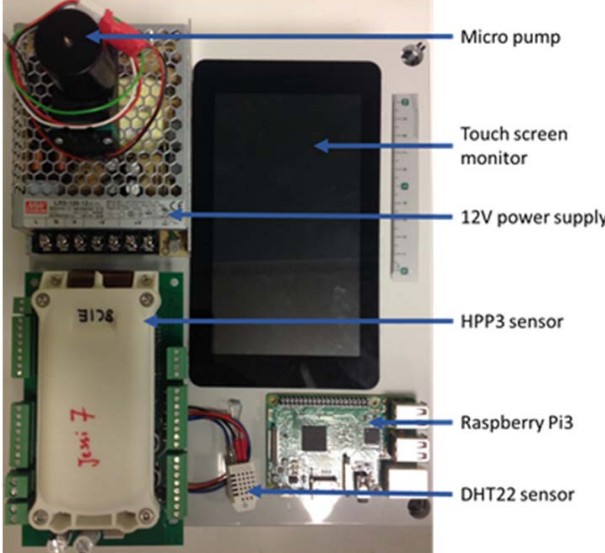

***Figure 1:*** Components of the portable instrument on the top of its box.
**3. Methods**



NDIR sensors are sensitive to IR light absorption by $CO_2$ in the air contained in their optical cell,
but the retrieval of $CO_2$ concentration to the desired accuracy of 1 ppm is made difficult by
sensitivities to temperature, pressure and humidity. Therefore, these parameters should be
controlled as much as possible, and their sensitivities characterized, to calculate $CO_2$. A series
of tests were carried out to characterize the HPP3.1 (S1.1, S1.2, S1.3) and HPP3.2 (S2.1, S2.2,
S2.3) performances and sensitivities to $CO_2$, T, P and RH. Firstly, temperature, pressure and
$CO_2$ sensitivities were determined in laboratory experiments. Then, field measurements were
conducted with an accurate instrument (Picarro, USA, G2401) measuring the same air than the
HPP; The G2401 accuracy is estimated to be below 0.05 ppm (Rella et al., 2013). Table1
summarizes all laboratory tests and field tests measurements, which are presented upon in this
section. The general test setup for the experiments can be seen in Figure 3.

| Name | Purpose | Location | Air measured | Parameter | Range of T (°C) and P (atm) | Range of [CO2] in air (ppm) | Range of [CO2] in Cal. Cylinders (ppm) | Duration (days) | Sensors tested |
|---|---|---|---|---|---|---|---|---|---|
| **PT1** | Correlation between [CO2] and P / T | plastic chamber (Saclay) | Calibration cylinders | T, P | 16-32 and 0.965-1.025 | N/A | 420 to 450 | 3 | S1.1, S1.2, S1.3 |
| **PT2** | Correlation between [CO2] and P / T | PIT climate chamber (Guyancourt) | Calibration cylinders | T, P | -2 to 35 and 0.75 to 0.95 | N/A | 420 to 450 | 5 | S2.1, S2,2, S2.3 |
| **DA1** | Test calibration frequency | Laboratory (Saclay) | Calibration cylinders and dried outside air | T, P, H2O, [CO2] from CRDS | 24 to 31 and 0.996 to 1.010 | 417 to 575 | 330 to 1000 | 48 | S1.1, S1.2, S1.3 |
| **WA2-1** | Test calibration frequency | Laboratory (Saclay) | Outside air | T, P, H2O, [CO2] from CRDS | 25 to 27 and 0.999 to 1.008 | 389 to 508 | N/A | 45 | S2.2, S2.3 |
| **WA2-2** | Test calibration frequency | Laboratory (Jussieu) | Outside air | T, P, H2O, [CO2] from CRDS | 29 to 31 and 0.993 to 1.021 | 393 to 521 | N/A | 60 | S2.1 |

*Table 1:* Summary of all laboratory tests.
**3.1. Laboratory tests**
***3.1.1. Sensitivity to temperature and pressure variations***
Two series of temperature and pressure sensitivity tests (PT1, PT2) were realized in a closed
chamber with controlled T and P for the HPP3.1 and HPP3.2 sensors. These tests are for
assessing the linearity of the response of each sensor to $CO_2$ for different pressure and
temperature conditions. For the HPP3.1 sensors, an internal pressure compensation does not
exist, but the HPP3.2 series includes a pressure sensor together with a compensation algorithm
which normalizes measured $CO_2$ mixing ratios according to ambient pressure (Gaynullin et al.
202   2016).
In test PT1 (table1), three HPP3.1 sensors were put in a simple plastic chamber and exposed to
pressure changes ranging from 0.965 to 1.025 atm, and temperature ranges of 16 to 32 °C.
Pressure and temperature were measured by a high-precision pressure sensor (Keller,
Germany, Series 33x, 0.01% precision).





In test PT2, three HPP3.2 sensors were placed inside a dedicated temperature and pressure chamber at the Plateforme d'Integration et de Tests at OVSQ Guyancourt, France (PIT) where a much larger range of T and P variations could be applied. During each T, P test, four calibration cylinders with dry air $CO_2$ mixing ratios from 420 to 450 ppm were measured by all the HPP3.2 sensors for a period of approximately 120 hours for each cylinder. In the PIT chamber, temperatures was varied from -2 to 35 °C with linear rates of change of 1 °C/hour keeping pressure constant at a value of 0.95 atm. Pressure was varied from 0.95 to 0.75 atm with an increment of $5.10^{-2}$ atm, being regulated with a primary pump, with temperature fixed at 15°C.

### 3.1.2. Calibration of $CO_2$ variations measured by the sensors for dry and wet air

These experiments were performed to evaluate the response of HPP sensors to $CO_2$ changes in ambient air. Two modes of operations have been tested i.e. using either a dried or an undried gas streams, described as follows.

**Dry air experiments**

Water vapor is known to interfere with $CO_2$ measurements, in particular for NDIR sensors. It is thus important to determine the sensitivity of the sensors to $CO_2$ under the best possible conditions, that is, dry air. In test DA1 (Table 2) different HPP sensors were flushed with the same dry ambient air, passed through a Nafion dryer (PermaPure, USA, Model MD-070). Online CRDS measurement confirmed that $H_2O$ was reduced to trace amounts, i.e. 0.05 ±0.05 % $H_2O$. HPP3.1 sensors S1.1, S1.2, S1.3 were tested extensively during 45 days, and HPP3.2 sensors S2.1, S2.2 were tested during 12 days. No significant difference was found between the performances of both HPP versions. The experimental setup is shown in Figure 2. For the period of 45 days during spring 2016, S1.1, S1.2 and S1.3 measured dry ambient air in parallel with a co-deployed CRDS instrument (Picarro, USA, G2401). The ambient air was pumped from a sampling line fixed on the roof of the building (ca. 4 m a.g.l.) to flush the setup described in Fig. 2. In addition, four dry-air calibration cylinders (330 ppm, 375 ppm, 445 ppm and 1000 ppm of $CO_2$) were measured successively each 13 hours during 30 minutes (Figure 3). As the HPP responses can be slow, in order to remove memory effects, only the last 15 minutes of each calibration periods were used.

**Wet air experiments**

Because in the field, drying is often impractical, we also measured the HPP sensitivities to water vapor under worst-case conditions, that is both ambient air and calibration cylinders air being wet. If these sensitivities were stable over time, they could be used to correct $CO_2$ for the $H_2O$ interference. For WA2-1 and WA2-2 tests, the Nafion dryer was removed from the setup.



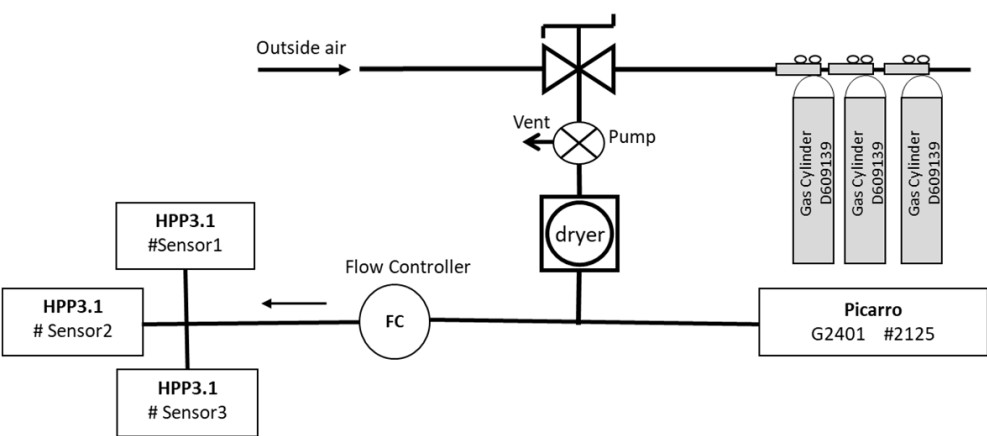

**Figure 2:** HPP3 instruments test Setup

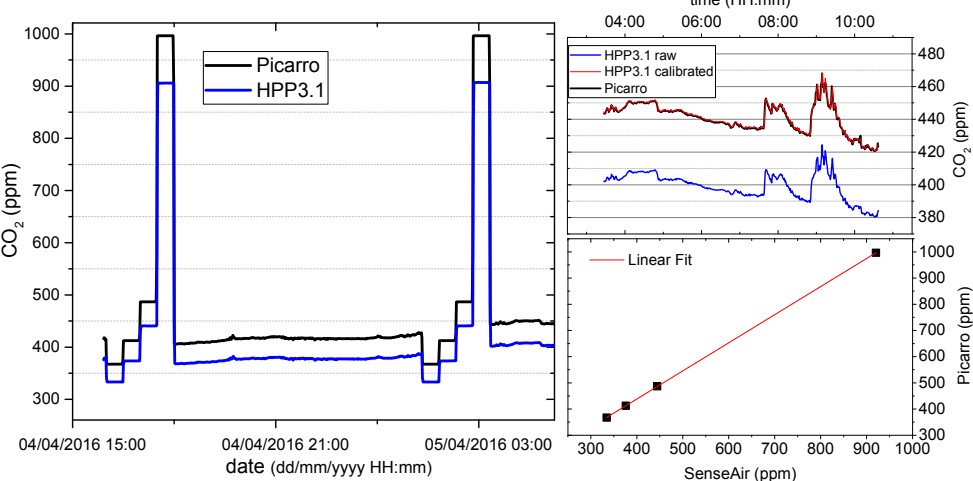

**Figure 3:** Left – $CO_2$ mixing ratios measured by S1.1 (blue) and the Picarro CRDS analyzer (black). Right - calibrated mixing ratios of S1.1 (red) compared to the raw values (blue). Below: Four reference gases (367 ppm, 413 ppm, 487 ppm and 997 ppm of $CO_2$), are used for the calibration. No saturation effects are observed within our $CO_2$ mixing ratio range.

**3.2. Instrument calibration procedure**

**Dry air calibration**

For dry air measurements in test DA1, a linear calibration curve is used. The lower right graph of Figure 3 shows that the response of the HPP3 instruments to $CO_2$ mixing ratio is linear ($R^2$ =



0.95) from 330 to 1000 ppm. No saturation effects are observed within this $CO_2$ mixing ratio
range since residuals are included in the ±1 ppm range.
**Wet air calibration**
Due to the high correlation of air temperature and water vapor content, we have applied a
multivariate regression method, which includes all environmental variables. Indeed, if variables
are corrected one at a time, an overcorrection of one of the correlated variables may occur.
Multi-linear regression is a generalization of linear regression by considering more than one
variable. We used a multivariate linear regression of the form:
$y = b + a_C x_C + a_p x_P + a_T x_T + a_w x_w + a_{xy} x_{xy}$     (1)
$y$ is the $CO_2$ value measured by the reference accurate instrument (CRDS), considered as the
true value here. $X_C$ is the $CO_2$ measured with the HPP3 instrument, with additional factors to
capture the influence of the pressure $P$, the temperature $T$, the water mixing ratio $W$, the
baseline drift $xy$ and a baseline offset $b$. Baseline drifts are corrected by fitting a linear y=x curve
in which y is incremented with time. Instrument specific coefficients for the multivariate linear
regression are determined using measurements of the parameters during several days.
**3.3. Field tests with urban air measurements**
To assess their real-world performance of the sensors, we conducted additional tests for the
HPP sensors measuring outside air under typical conditions for urban air monitoring, after the
sensors were fully integrated into instruments as described in section 2.2. Three HPP3.1
instruments (S1.1, S1.2, S1.3) and two HPP3.2 instruments (S2.2, S2.3) were installed in the
laboratory building of Saclay (48.7120N, 2.1462E) to measure outside air on top of the building
roof. Saclay is located 20 kilometers south of the center of Paris in a low-urbanized area. In
addition, one HPP3.2 instrument (S2.1) was installed to measure air on the Jussieu University
campus in the center of Paris (48.8464N, 2.3566E).
*3.3.1. Saclay Site*
The sampling line, a 5 meter Dekabon tube with an inner tube of 0.6 cm, was fixed on the
rooftop of the building at about 4 meters above the ground. The laboratory hosting the five
instruments is equipped with a cooling/heating unit that was turned off most of the time so that
room temperature varied between 24 and 31 °C. A CRDS (Picarro, USA, G2401) measured air
from the same intake in parallel as seen in Figure 3. A diaphragm pump (KNF Lab, Germany,
Model N86KN.18) was used to pump air to the five instruments, and a flow controller
(Bronkhorst, France, El-Flow series) was used to regulate the air-flow distributed with a manifold
to the HPP3 instruments at 500 mln min$^{-1}$ to ensure stable experimental conditions. For dry air
measurements using the HPP3.1 (48 days) an external Nafion dryer (PermaPure, USA, MD-070
series) was used to eliminate $H_2O$ traces in the gas line during dry air experiments, while
HPP3.2 were tested for 45 days. Four reference gas cylinders (330ppm, 375ppm, 445ppm and
1000 ppm of $CO_2$) were used and each HPP was flushed every 12 hours for 30 minutes during
the dry air experiment. No calibration cylinders were used during the undried air experiment,
since the calibration was based on the co-located high precision measurement with the CRDS
analyzer. The mean mixing ratio of ambient $CO_2$ was 420 ppm and varied between 388 ppm and
575 ppm during dry air experiments and a mean of 409 ppm and varied between 389 ppm and
509 ppm during undried air experiments.



### 3.3.2. Jussieu site

The measurements from the HPP3.2 instrument (S2.1) in Jussieu were compared with those of
a co-located CRDS (Picarro, USA, G2401). Two independent sampling lines (about 5 meter
Dekabon tube with an inner tube of 0.6 cm) were used for the Picarro, and the S2.1 instrument.
The air-flow into HPP3.2 instrument was regulated by the micro pump (see section 2.2) and set
to 500 mln min$^{-1}$. At this location neither calibration cylinders nor a drying system were deployed
for the HPP3.2. The measurement period was 60 days and the mean ambient $CO_2$ mixing ratio
was 410 ppm and minute averages varying between 393 ppm and 521 ppm. Room temperature
varied between 28 and 31 °C during the observation period.

**4. Results**
### 4.1. Sensitivity to temperature and pressure variations using dried air
### 4.1.1. HPP3.1 instruments tested in the simple chamber
As shown in Figures 6 and 7 linear relationships are observed between $CO_2$ concentrations and
P, T ($R^2$ =0.99 with P and $R^2$ = 0.92 with T) in the simple chamber. Due to the limitation of
experimental conditions in these simple plastic chambers, only a narrow pressure range of 0.965
to 1.020 atm and a temperature range of 16 °C to 32 °C could be tested for these instruments.
Different slopes and intercepts are found for each instrument as reported in Table 2. This
indicates that there is no single universal P and T calibration curve that could be determined for
one instrument and used for others.
### 4.1.2. HPP3.2 instruments tested in the PIT chamber
The PIT tests results with P changes from 1.00 to 0.75 atm with an increment of $5.10^{-2}$ atm are
shown in Figure 4. The top panel of the Figure 4 shows the variations of $CO_2$ mixing ratios due to
P (from 0.5 to 1.8 ppm per 0.1 atm). Despite the built-in pressure compensation algorithm
developed for HPP3.2, $CO_2$ and P can still co-vary with a positive (S2.1 and S2.2) or a negative
(S2.3) correlation, indicating that an additional correction would be required when aiming to
achieve the best possible results. Consequently, we applied a linear fit between $\Delta CO_2$
(differences between the known mixing ratio in the cylinder and the mixing ratio measured by
HPP3.2 instruments) and pressure (Figure 6). The slope and intercept obtained are then used to
determine the offset due to P variations that has to be added on raw $CO_2$ mixing ratios reported
by the HPP3.2 instruments. The lower panel of Figure 4 shows that the corrected $CO_2$ mixing
ratios values have a root mean square deviation from the true mixing ratio in the calibration
cylinder (428.6 ppm) of less than 0.02 ppm for all three HPP3.2.
Figure 5 shows the effect of temperature variations in the PIT chamber going from -2 to 35 °C
(see section 3.1) on raw reported $CO_2$ mixing ratio of the HPP3.2 instruments. Temperature
corrections are done following the same steps as for P. For the three HPP3.2, $CO_2$ mixing ratios
are negatively correlated to T. As for the tests in the simple chamber with the HPP3.1
instruments, different linear T slopes and intercepts are observed for each HPP3.2 instrument
(Figure 6) in the PIT chamber. After correction for temperature variations, we obtain corrected





$CO_2$ mixing ratio values with a root mean square deviation which does not exceed 0.01 ppm
from the true value of the cylinder ($[CO_2]$=444 ppm) for the three HPP3.2 instruments.

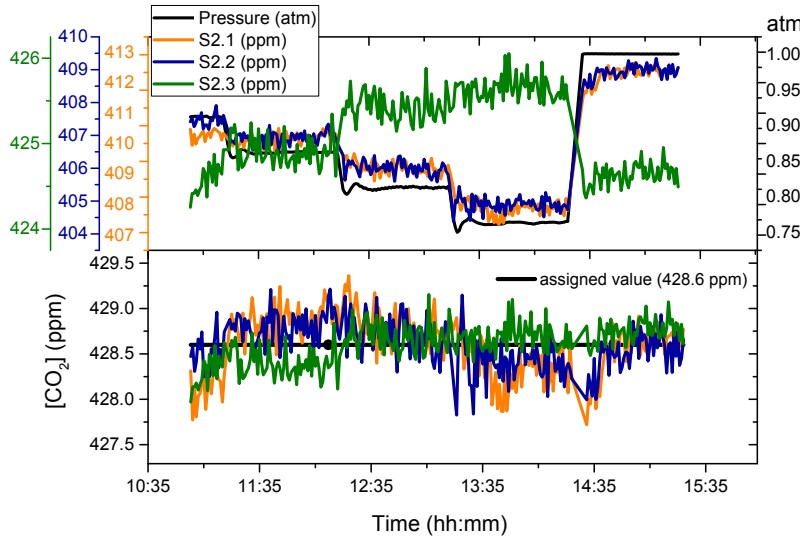

**Figure 4:** Upper panel – Effect of pressure variations (black) on raw $CO_2$ mixing ratios of three
HPP3.2 instruments measuring $CO_2$ from air from the same calibration cylinder (true value
=428.6 ppm), S2.1 (orange), S2.2 (blue) and S2.3 (green), please not the different y-axis scales.
Lower panel – Corrected $CO_2$ mixing ratios for HPP3.2 instruments.

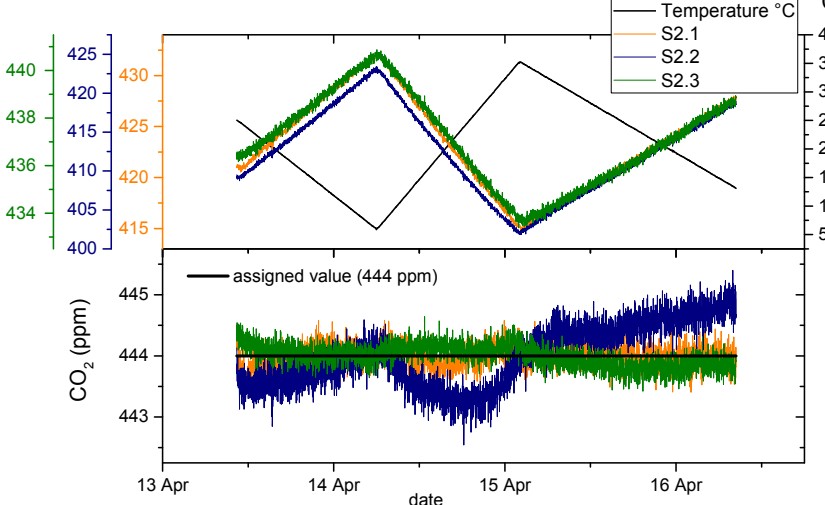






**Figure 5:** Upper panel – The effect of temperature variations (black) on raw $CO_2$ mixing ratios of HPP3.2 instruments measuring $CO_2$ from air from the same calibration cylinder (true value =444 ppm), S2.1 (orange), S2.2 (blue) and S2.3 (green), please not the different y-axis scales. Lower panel – Corrected $CO_2$ mixing ratios for HPP3.2 instruments.

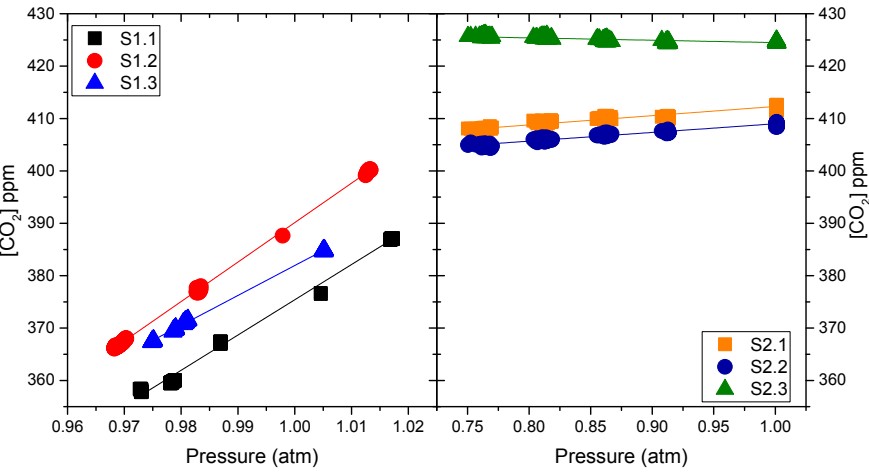

**Figure 6:** Linear relationship experimentally found between $CO_2$ mixing ratios and P for the HPP3.1 instruments (S1.1, S1.2, S1.3) (left) and for the HPP3.2 instruments (S2.1, S2.2, S2.3) (right). Note the different P range, going from 0.96 to 1.02 atm. for the HPP3.1 in the simple plastic chamber to 0.75 to 1 atm. for the HPP3.2 in the PIT chamber.

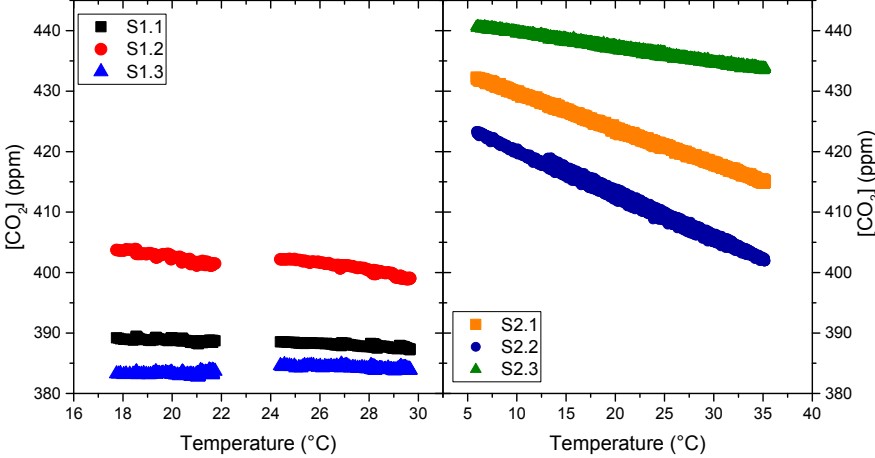

**Figure 7:** Linear relationships between $CO_2$ mixing ratios for HPP3.1 S1.1, S1.2 and S1.3 (left)



at temperature values going from 17 to 30 °C in plastic chamber, and for HPP3.2 S2.1, S2.2 and
S2.3 at temperature values going from 5 to 35 °C in the PIT chamber.
Table 2 summarizes the results of the pressure and temperature tests for all the HPP3
instruments. These tests results show a sensor-specific response to P and T. A large difference
of $CO_2$ mixing ratios sensitivity to pressure variations is observed between the two HPP3
versions. A sensitivity of 571.5 to 753.4 ppm/atm is found for the HPP3.1 versions, whereas this
sensitivity ranges from -4.5 up to 17.6 ppm/atm for the newest HPP3.2 versions. The lower
sensitivity among HPP3.2 prototypes is due to the pressure compensation algorithm which is
included in this model. Since the pressure compensation algorithm does still not fully correct the
$CO_2$ variations due to pressure changes, we found that it is necessary to apply a correction for
pressure interferences on the $CO_2$ mixing ratios signal, and that is correction should be sensor
specific. The $CO_2$ mixing ratios sensitivity to temperature variations are found to be in similar
ranges for both sensors. Sensitivities of -0.3 to 0.1 ppm/°C and -0.2 to -0.7 ppm/°C are found for
HPP3.1 and HPP3.2 prototypes respectively.

|  | Pressure | | | Temperature | | |
|---|---|---|---|---|---|---|
|  | Slope (ppm/atm) | Intercept (ppm) | $R^2$ | Slope (ppm/°C) | Intercept (ppm) | $R^2$ |
| S1.1 | 673.1 ±4.4 | -297.7 ±4.3 | 0.94 | -0.124 ±0.003 | 391.34 ±0.07 | 0.85 |
| S1.2 | 753.4 ±1.1 | -363.3 ±1.1 | 0.95 | -0.29 ±0.01 | 408.1 ±0.2 | 0.80 |
| S1.3 | 571.5 ±1.4 | -189.5 ±1.4 | 0.94 | 0.107 ±0.004 | 381.2 ±0.1 | 0.63 |
| S2.1 | 17.6 ±0.2 | 394. ±0.2 | 0.95 | -0.5854 ±0.0004 | 435.530 ±0.01 | 0.99 |
| S2.2 | 16.6 ±0.2 | 392.4 ±0.2 | 0.97 | -0.716 ±0.001 | 427.31 ±0.02 | 0.99 |
| S2.3 | -4.5 ±0.0 | 429.0 ±0.0 | 0.75 | -0.2453 ±0.0004 | 442.16 ±0.01 | 0.99 |

**Table 2:** Slopes and intercept calculated for $CO_2$ correction due to temperature and pressure.
Sensor 1 to 3 are type HPP3.1, whereas Sensor 4 to 6 are HPP3.2.
From Figures 6 and Table 2, we see a positive correlation between $CO_2$ and P for five
instruments (S1.1, S1.2, S1.3, S2.1 and S2.2) and a negative correlation for S2.3. In Figures 7
and Table 2, a negative correlation between $CO_2$ and temperature is found for 5 instruments
(S1.1, S1.2, S2.1, S2.2 and S2.3) and a positive one for S1.3. After correcting for temperature
and pressure, no more correlations are observed between corrected $CO_2$ and pressure and
temperature. Corrected $CO_2$ mixing ratios of HPP3.2 are stable and standard errors do not
exceed 0.3 ppm and 0.2 ppm for pressure and temperature corrections respectively, except for





$CO_2$ mixing ratios after temperature correction for S2.2 which reaches a standard deviation
(STD) of 0.5 ppm. However, we do not reach the same stability after pressure and temperature
correction for HPP3.1 prototypes. Standard deviations of 0.9, 0.2 and 0.2 ppm are calculated for
S1.1, S1.2 and S1.3 respectively after pressure correction, and Standard deviations of 1.3, 2.6
and 1.6 ppm are determined for S1.1, S1.2 and S1.3 respectively after temperature corrections.
These differences between the results of the two HPP3 versions can be explained by the fact
that HPP3.2 prototypes had the opportunity to be tested in a sophisticated climatic chamber
which respects precise temperature and pressure setpoints and in which only one of the two
variables are modified one at a time.

**4.2. Instrument calibration and stability during continuous measurements**

Given that the instrument response to $CO_2$ is affected by atmospheric water vapor, we present
the results from dried and wet ambient air measurements separately.

***4.2.1. Measurements of dried ambient air (test DA1)***

Four calibration cylinders were used in order to linearly calibrate the three HPP3.1 instruments
(see section 3.1). To assess the quality of this calibration, the RMS relative to 1-minute $CO_2$
mixing ratios from the co-located CRDS data were calculated, and shown in Figure 8. Although
calibration cylinders were measured each 12 hours, by ignoring deliberately some calibrations,
we processed the time series to re-compute $CO_2$ assuming a range of different time intervals
between two calibrations. The results shown in figure 8 are for calibrations intervals of 0.5, 6, 12,
19, 25, 31, 38 and 45 days. Each point in this Figure corresponds to the RMS values calculated
for the HPP3.1 instruments S1.1, S1.2, S1.3.
We find that the 1 ppm accuracy threshold is met (and even surpassed) when measuring dried
air and for calibration intervals no longer than 6 days. We also see a marked difference between
the performances of each sensor: S1.1 shows the best performance, followed by S1.3 and S1.2.
The larger mean variation of $\Delta CO_2$ (±4 ppm for S1) are observed for 19 days between two
calibrations. Surprisingly, one calibration each 45 days does not deteriorate the mean of $\Delta CO_2$
significantly. Indeed, the variability of mean $\Delta CO_2$ decreases over longer time periods as the
instruments do not have a residual no persistent long-term drift and positive and negative values
of $\Delta CO_2$ cancel each other over time. The RMS of the minute averages slowly increases with
increasing calibration intervals but seems to stabilize between 3 and 4 ppm.




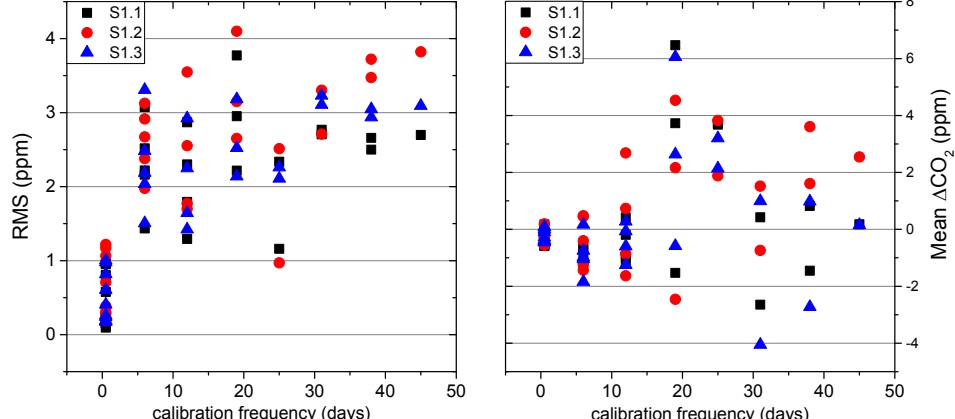

***Figure 8:*** RMS (left) and mean difference (right) of $CO_2$ of the three HPP3.1 instruments
compared to an independent accurate CRDS Picarrro, during a measurement period of 48 days.
The x-axis represents different chosen frequencies for calibration of the HPP3.1 using reference
cylinders.

***4.2.2. Saclay ambient air measurements (test WA2-1)***
During this test (section 3.3, *Saclay site*), all atmospheric variables in wet air which affect the
performance of the instruments, i.e. pressure, temperature, water vapor content and $CO_2$ mixing
ratios, were measured. As described in section 3.1, a six term multivariate linear regression is
used to calculate the regression coefficients for each HPP3.2 instruments. Panel of Figure 9
shows the results for measurements from July 20[th] until August 8[th] 2016 of ambient air at Saclay
from instrument HPP3.2 S2.2.
To illustrate the relative impact of the sensitivity to each variable on the reported raw data of $CO_2$
measurements each component of the multi-linear fit is added separately (one at a time here).
Overall, we show 5 correction variables starting from the offset correction, in which only the
offset of the regression is corrected until to the last panel in which all five terms of equation 1 are
taken into account. The offset and concentration dependent corrections terms (b and $a_c x_c$) are
the most significant corrections among all 5 parameters and allow reducing the mean $\Delta CO_2$ from
45 ppm to 0 ppm. The other 4 parameters (pressure, temperature, water vapor and drift
corrections) further reduce the difference between CRDS and HPP3.2 reducing the RMS of
minute averages from 1.03 ppm to 0.67 ppm. Here, the temperature correction (d) and the water
vapor correction (e) provide a correction of similar magnitude, keeping the same RMS and
improving mean $\Delta CO_2$ only from 0.16 to 0.13 ppm. This is understandable since temperature
and water vapor are correlated for this type of measurement.





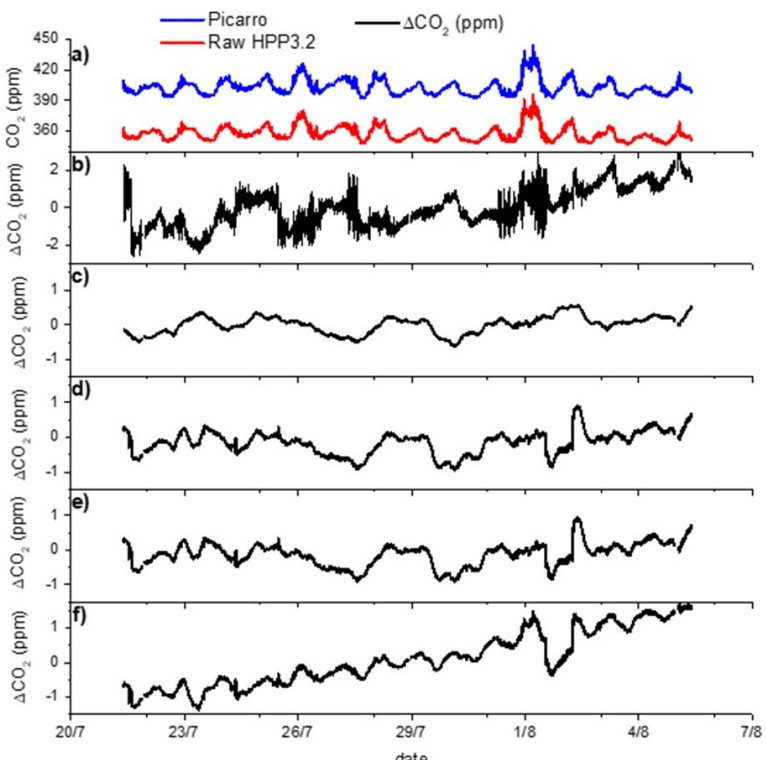


**Figure 9:** A continuous time series of 1 min averages for HPP3.2 instrument S2.2 compared to the Picarro CRDS instrument after correcting for the different variables for a period of 15 days. Plot (a) shows raw $CO_2$ measured by S2.2 and the Picarro. Plot (b) shows the difference between Picarro and S2.2 after offset correction. The next 4 plots (c), (d), (e), (f) show the difference to plot (b) after having correcting the HPP3.2 $CO_2$ to fit the Picarro $CO_2$ using pressure, temperature, water vapor, and linear drift respectively. RMS and mean values of $\Delta CO_2$ ($[CO_2]_{picarro}$ - $[CO_2]_{S2.2}$) data after each correction are shown in Table 3.


|  | raw | Offset correction | Pressure correction | Temperature correction | RH correction | Drift correction |
|---|---|---|---|---|---|---|
| RMS (ppm) | 1.11 | 1.03 | 1.00 | 0.97 | 0.97 | 0.67 |
| Mean (ppm) | 45.33 | 1.0 10-3 | 8.6 10-4 | 0.16 | 0.13 | -0.08 |



***Table 3***: RMS and mean values of one minute average $\Delta CO_2$ ($[CO_2]_{picarro}$ - $[CO_2]_{S2.2}$) data for each
correction step. Note that corrections are cumulative from left to right.
A second instrument was deployed but the acquisition failed for a period of one week leading to
a discontinuity in the data.
The laboratory studies (section 4.2.2.) already indicated that recalibration of the HPPs is
required because of sensitivities to T, P and water vapor that are instrument-specific. We call the
period during which the six calibration coefficients of Eq. (1) are calculated by fitting the PIcarro
$CO_2$ time series, the learning period. Attempting to determine those calibration coefficients
during a short learning period e.g. of one week, leads to high mean $\Delta CO_2$, as can be seen in
Figure 10. A learning period of two weeks leads to significantly better results. We tested
systematically longer learning periods, of up to 45 days. The raw measurement data not used in
the learning period is calibrated and compared to the CRDS system which then aids as a tool to
assess the performance of this $CO_2$ calibration approach.
We also compared different learning periods of the same length. As an example, considering a
45 days experiment, we chose 3 different learning periods of successive 15 days. We also
tested the approach of using the first and last weeks of a 45 days period to create a non-
successive two weeks learning period.
Figure 10 shows the RMS and mean $\Delta CO_2$ values considering 3 learning periods (C1, C2, C3)
of 15 days each. The regression coefficients of the multilinear model of Eq. (1) for C1, C2 and
C3 are calculated using the first, second and third consecutive 15 days of the experimental
period. These coefficients are then used to predict corrected $[CO_2]_{HPP3.2}$ for the three cross-
validation periods of 15, 30 and 45 days. Also, calibration coefficients (W1, W6) were calculated
using the first and sixth week of the 45 days period for learning. Unsurprisingly, using C1
coefficients gives the best results for the first 15 days used for training (RMS=0.6 ppm and
mean=-0.1 ppm for hourly values), and lead a higher bias for the last 15 days (RMS=1 ppm and
mean=-1 ppm for hourly values). Using C2 coefficients to correct adjacent 15 days from the
learning period gives comparable results (RMS=0.7, mean=-0.6 ppm and RMS=0.7, mean=-0.1
ppm respectively for the first and last 15 days). Considering the last learning period, C3
coefficients show a mean bias of -2.5 ppm when learning is from the first 15 days. One reason
that can explain this behavior is the greater variability of $CO_2$ mixing ratio during the last 15 days
of the experiment. The interquartile range of $CO_2$ mixing ratio is 10, 15 and 25 ppm respectively
for the first, second and third period. The CO2 mixing ratio correction is accomplished mostly by
correcting T, P, $H_2O$ and the instrument offset. A small variation of sensitivities may lead to a
less appropriate correction for periods of smaller variability. Another reason for this difference is
the drift component of the correction in Eq. 1. The linear drift of the instrument also varies with
time. One method to better correct for the slow linear drift of the instrument is to combine the first
and last week of the experiment into a learning period instead of using two consecutive weeks.
Figure 11 shows corrected $\Delta CO_2$ ($[CO_2]_{picarro}$ - $[CO_2]_{HPP3.2}$) of S2.2 during 45 days when using this
approach. When using the first week (W1) and the last week (W6) for learning, the instrument
drift is not properly corrected and a residual slope of 0.14 and 0.28 ppm/week is shown in the
black (W1) and the red (W6) curves of the figure, respectively. Nearly no drift (0.01 ppm/week) is
observed when considering both W1 and W6 for the training (blue curve). On Figure 10,
magenta stars show RMS and mean $\Delta CO_2$ values of the whole 45 day time series considering
both W1 and W6 as learning periods. With this coefficient determination method, mean $\Delta CO_2$
bias can be reduced to nearly 0 ppm. Finally, averaging the 1-minute HPP3.2 data to hourly
averages can further improve RMS values up to 28%. As expected, mean values do not change
for hourly averages.




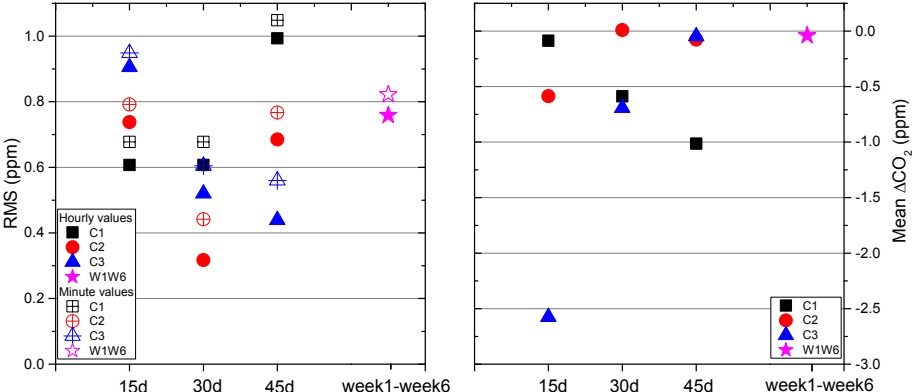

***Figure 10:*** left – RMS values considering 3 learning periods of 15 consecutive days each in x-
axis. C1, C2 and C3 correspond to RMS calculated considering correction coefficients
determined from learning, that is, fitting of Eq. 1 to Picarro data during the first, second and third
15 consecutive days respectively. Week1-Week6 corresponds to RMS calculated considering
correction coefficients determined during the first and last weeks of the experiment. Hourly and
minute values are represented in full and empty symbols respectively. Right – Mean $\Delta CO_2$
calculated for the four learning periods choices.

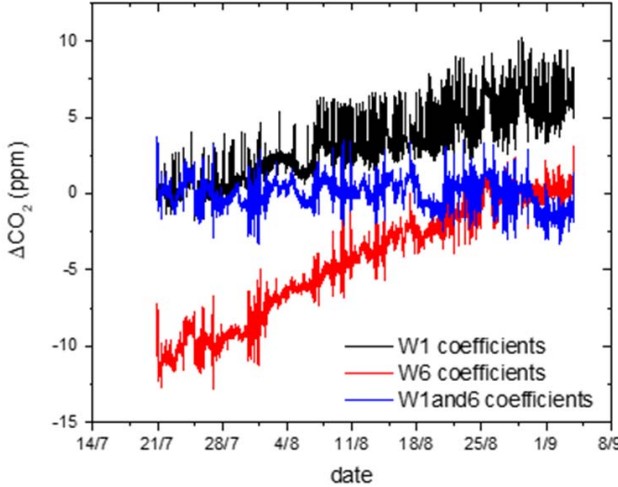


***Figure 11:*** $\Delta CO_2$ ([$CO_2$]$_{picarro}$ - [$CO_2$]$_{HPP3.2}$) of HPP3.2 instrument S2.2 during 45 days considering
different learning periods of one week. Results from learning periods of week one (W1) and





week six (W6) are in black and red respectively. The blue curve shows corrected $\Delta CO_2$ when both W1 and W6 are used in the learning.

### 4.2.3. Urban site of Jussieu (test WA2-2)

To assess the performance of the HPP3.2 instruments, wet ambient air measurements at the urban site of Jussieu were carried for 60 consecutive days using HPP3.2 instrument S2.1 alongside a Picarro upon which learning is applied. Figure 12 shows RMS and mean values calculated with four learning periods of 15 consecutive days each and one learning considering both first and last week of the experiment. Calibration coefficients for C1, C2, C3 and C4 are calculated considering learning periods of first, second, third and fourth 15 consecutive days of the experiment respectively. W1W8 coefficients are calculated considering week one (W1) and week eight (W8) of the experiment.

First, we look at the results using C1, C2, C3 and C4 coefficients. Out of the four consecutive 15 day learning periods, C1 coefficients seem to provide the best correction of raw $CO_2$ mixing ratio with hourly RMS values between 0.3 and 0.6 ppm and mean values between 0 and 1 ppm. Absolute mean values of $\Delta CO_2$ for C3 and C4 show a linear increase (slopes of 1.3 ppm per 15 days for C3 and 1.9 ppm per 15 days for C4) the further we go from learning periods leading to hourly mean values of -3.4 and -5.4 ppm respectively for C3 and C4 corrections. This is a typical case where the drift component could not be well characterized by the chosen learning periods. Another interesting observation concerns minute RMS values of C3 and C4 corrections which are lower than hourly RMS values for the same coefficients calculated during the first and second 15 days. Minute RMS are 0.1 and 0.4 ppm lower respectively for the first and second 15 day period for both C3 and C4 corrections. As for the previous urban measurements, considering both first and last weeks as a learning period provides satisfying results (RMS=1.1 ppm, mean=0.9 ppm) for the correction of raw $CO_2$ mixing ratios during a period of a month and a half.

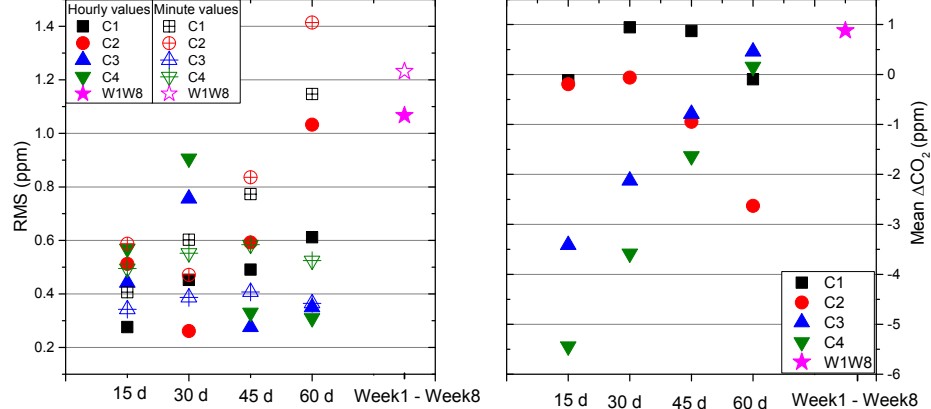

**Figure 12:** left – RMS values considering 4 learning periods of 15 consecutive days each. C1, C2, C3 and C4 corresponds to RMS calculated considering correction coefficients determined during the first, second and third 15 consecutive days respectively. W1W8 correspond to RMS calculated considering correction coefficients determined during the first and eighth weeks of the



experiment. Hourly and minute values are represented in full and empty symbols respectively.
Right – Mean $\Delta CO_2$ calculated with the same five coefficients mentioned above.

**5. Conclusion and perspective**

We integrated HPP3.1 and HPP3.2 NDIR sensors into a portable low-cost instrument with
additional sensors and internal data acquisition. The laboratory tests reveal a strong sensitivity
of measured $CO_2$ mixing ratios to ambient air pressure for the HPP3.1 series and a significantly
decreased sensitivity to pressure, even for the upgraded HPP3.2 sensors equipped with a P
sensor and using a manufacturer P-correction. To achieve the required stability and accuracy for
urban observations, instruments have to be corrected at regular intervals against data from a
very accurate reference instrument to account for their cross-sensitivities to T, P, $H_2O$ changes
and electronic drift, unless those parameters could be controlled externally in the future. We
found that commercially available P, T and RH sensors that are compatible with the chosen
Raspberry Pi3 platform are sufficiently precise to use these parameters as predictors of the
linear equation use to calibrate each HPP instrument against the very accurate reference
instrument, a process called learning.
Two common modes of operations have been successfully tested i.e. using the low-cost medium
precision instrument for either dried or undried gas streams. Our results indicate that using a
dried gas stream does not improve measurement precision or stability compared to an undried
gas streams provided that a multilinear regression model is used for calibration (learning), which
accounts for all cross-sensitivities including to $H_2O$ mixing ratio changes.
We furthermore find that sensor specific corrections are required and they should be considered
time-dependent, e.g. by including a linear drift that only becomes more apparent for longer-term
observations. Different calibration strategies were tested for the Saclay and Jussieu ambient air
measurements based on reference CRDS systems, and their results evaluated against CRDS
cross validation data that were not used for learning. Those sites exhibit the typical mixing ratio
enhancement in urban GHG monitoring networks were LCMP instruments could be deployed in
the future. Regular (6 weekly) re-calibrations are found to be appropriate to capture sensor linear
drifts and changes in relevant cross-sensitivities, while not increasing the burden of performing
calibration too often by transplanting the low cost instrument to measure CO2 in parallel with a
CRDS. Learning periods of one week with parallel CRDS measurements, spaced by a 'free
running' period of 45 days, was sufficient for the HPP data to be within 1 ppm of CRDS during
that period. This calibration approach by learning can be an alternative to permanently deploying
calibration gases for each individual sensor. Overall, the requirement of ca. 1ppm compatibility
for hourly means $CO_2$ mixing ratios for a dense $CO_2$ monitoring network in Paris (Wu et al. 2016)
was achieved and no significant long-term bias was detected.
The field tests at the Saclay and Jussieu station are being continued to see if the instrument
performance deteriorates over its lifetime. Since the start of the test in 2015 until now multiple
HPP3.1 sensors have been in use for without significant performance loss.
Future improvements for the LCMP instruments will include the addition of batteries to allow their
transport to the central calibration lab without power cut as well as using them in field
campaigns, e.g. landfills when connected to solar panels or small wind turbines. During future
tests at sites without reference instruments, small pressurized gas containers (12l, minican,
Linde Gas) will be used to regularly inject target gas to track the performance.
The overall operational cost of the new calibration scheme using a central laboratory and
rotating the LCMP systems can also only be assessed after more extensive field deployment
has been performed.





**Acknowledgements**
The work conducted here was partially funded through the LOCATION project of the Low Carbon City Laboratory and a SME-VOUCHER from climate-KIC (EIT) as well as the Chaire BridGES of UVSQ, CEA, Thales Alenia Space and Veolia S.A.

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
