# Peer review of "Manuscript under review for journal Atmos. Meas. Tech."

_Atmospheric Measurement Techniques, 2018_

## Referee Comment (RC1) · Anonymous Referee #1 · 31 Oct 2018

This paper reports a series of experiments evaluating one particular low cost CO2 sensor. The paper is not especially well organized. It reads as a long list of experiments. However, analysis that synthesizes the observations and context from other related work is lacking.

There is an extensive knowledge base of performance for such NDIR instruments, including manufacturer literature (e.g. LiCor, Vaisala), and field evaluation in other contexts.

Key ideas in the description of NDIR sensors that this paper could be better organized around include:

1) that they measure absorption which is proportional to number density but the atmospheric quantity of interest is dry air mixing ratio. The measured quantity must be converted using the ideal gas law and subtracting water number density to give the dry air mixing ratio. Many of the figures are some form of confusing intermediate product along the way to a dry air mixing ratio.

2) that a second order correction is associated with pressure broadening of the CO2 absorption lines.

3) that knowledge of zero is as challenging as knowledge of response to CO2.

The paper neglects to acknowledge or build on related work by Shusterman et al. Atmos. Chem. Phys., 16, 13449-13463, 2016 and Zimmerman et al. Atmos. Meas. Tech., 11, 291-313 2018 and likely others.

Overall I recommend a substantial revision to improve the clarity. Cutting the number of figures in half and targeting them to identified issues with performance would be welcome.

---

## Referee Comment (RC2) · Anonymous Referee #2 · 23 Nov 2018

**1   General comments**

This manuscript reports on newly developed commercial CO2 sensors which were tested in order to assess their suitability for deployment in urban monitoring networks. The sensitivity of the low-cost sensors to environmental factors is quantified and corrected by means of a multivariable regression. The topic is of high interest in the community and the presented work fits well the scope of AMT. The authors have carried out a series of experiments well suited to assess the accuracy that can be expected in

the proposed use case.

Unfortunately, the manuscript in its current state does not realise the full potential of the extensive data set collected. Structure, precision and clarity of the language as well as formal aspects need substantial improvements. In general I strongly recommend consultation of AMT's Manuscript preparation guidelines for authors, especially the sections on manuscript composition, mathematical notation and terminology as well as the English guidelines and house standards.

The structure of the manuscript suffers from loose connections between sections and a lack of coherence in the order in which information is presented. It is not easy to see which sections in the text and which figures belong together. This might be improved by referencing the short names of the corresponding experiments in every figure caption and at the beginning of each subsection. The explanation of the different experiments in Sect. 3 needs to be extended and improved. In the current manuscript, important details are missing or the reader has to combine bits of information from different sections, tables and figures to understand how the experiment was carried out. I suggest to give all descriptions a common structure, e.g. like this: (1) Purpose of the test (2) Which sensors were tested? (3) Which air was measured, i.e. ambient or from cylinder? Was the drier used? If cylinders were used, for how long and how often was switched between them? (4) Which pumps and means of flow control were used? (5) Was there a reference measurement by a CRDS analyser? If yes, how was it connected to the system? (6) What were the ambient conditions of the sensors? In case of controlled temperature and pressure, describe the pattern. Show a graph of the ambient conditions where they matter for the experiment. All in all, make sure that the reader gets all information necessary for repeating the experiment and that this information is provided in a single section.

Precision and clarity of the language is a big issue in the current form of the manuscript. Some passages are hard to understand (see Specific comments). In several places in the manuscript the value of a quantity is given without naming the quantity itself. To

avoid misinterpretation, name and value of a quantity should always be provided together, e.g. "an operating voltage of 12 V direct current". See the "International vocabulary of metrology – Basic and general concepts and associated terms (VIM)" (JCGM 200:2012) for details. AMT demands that "wherever possible, SI units should be used". Throughout the manuscript, "atm" should be replaced by "kPa". Other problematic units are listed under Specific comments and Technical corrections.

I strongly urge the authors to differentiate between the substance CO2 and its abundance. "CO2" cannot be the quantity plotted on the axis of a diagram. What is really meant is a measure for the abundance of CO2. The abundance can be expressed e.g. as a mole fraction (measured in ppm) or a concentration (measured in different units like g/L, mol/L), see e.g. the IUPAC Gold book. The expression "[A]" is commonly used for the concentration of substance A. The Picarro analysers report mole fraction, not concentration, so "[CO2] from CRDS" is misleading. Mixing ratio is yet another quantity (see WMO Guide to Meteorological Instruments and Methods of Observation). As far as I can see, there is no need to refer to concentrations or mixing ratios in this manuscript and all occurrences of these quantities can be adapted such that CO2 is quantified by its dry air mole fraction. Even when wet air is measured, the correction for the influence of water vapour leads to an estimation of dry air mole fraction.

One important factor that makes the text hard to follow is the lack of consistency in names and categories. To give an example of the problems with the current naming scheme: Sect. 3.1 is named "Laboratory tests", Sect. 3.3 is called "Field tests with urban air measurements", suggesting that laboratory and field tests are two different things. However, Table 1 has the caption "Summary of all laboratory tests", yet it contains also the measurements WA2-1 and WA2-2 which seem to be the experiments described in Sect. 3.3. Moreover, the location for WA2-1 and WA2-2 (field tests?) according to Table 2 is "Laboratory (Saclay)", while the location for PT1 (a laboratory test) is "plastic chamber (Saclay)".

Please decide for either of the terms "HPP sensors" or "HPP3 sensors" when referring

to all sensors that were tested. Mixing the terms confuses the reader. Likewise, please refer to specific sensors consistently with their short name defined in Sect. 2.1, e.g. "S2.2". Do not occasionally use "HPP3.2 S2.2" or similar constructs.

Calibration vs. correction: It would help the clarity if the act of determining correction coefficients was consistently referred to as "calibration" and the act of applying these correction coefficients to raw measurements was consistently referred to as "correction".

The abbreviation RMS for root mean square is used many times in the manuscript. As far as I can see, in all instances the root mean square difference or root mean square error is actually meant. It needs to be clear which quantities are subtracted to obtain this difference. To this end, I suggest a notation like "$\mathrm{RMS}(x_{\mathrm{S1.1}} - x_{\mathrm{CRDS}}) = 0.5\,\mathrm{ppm}$". Writing "$\mathrm{RMS} = 0.5\,\mathrm{ppm}$" is not acceptable.

As for the formal aspects I refer the authors to the comments below and AMT's guidelines for authors.

Note also that authors are requested to include a statement on how the data used in the work can be accessed by others (see AMT Data policy).

**2  Specific comments**

ll. 1–2: "using commercial NDIR sensors" should be left out or moved to a different position in the title. Suggestion: "Characterization of lower-cost commercial sensors for atmospheric carbon dioxide monitoring systems in urban areas"

ll. 36–39: sentence should be split. Furthermore, the second clause "[...] use networks [...] for [...] networks" is not very specific and should be reformulated. Suggestion: Replace "urban CO2 networks" with the higher goal, i.e. "monitoring of urban CO2 emissions".

l. 22: "of a HPP commercial NDIR sensors manufactured by Senseair AB " -> "of a newly developed sensors".

ll. 65–66: The meaning of the sentence is not clear, "only" seems to be in the wrong place. The cited reference does not include a case with four stations, please correct.

l. 99: "measure CO2 based on controlling parameters for ambient air" -> "measure the dry air mole fraction of CO2 in ambient air"

ll. 102-103: A reference instrument can be calibrated over a range, but a standard has just one value. Suggestion: "[...] a suite of gas standards with CO2 dry air mole fractions between 330 and 1000 ppm [...]"

ll. 137-139: What is the technical improvement of the sample cell redesign with respect to the topic of this publication, i.e. the use as a CO2 sensor? If there is none, please remove this point.

ll. 155–162: How where the in- and outlet of the pump, the HPP sensor, the pressure and temperature sensor and the air feed-through of the enclosure connected to each other? What is the response time of the sensors at the flow speeds used?

ll. 156–157: This sentence is redundant with ll. 142–143. Remove it.

ll. 157–159: Use SI units for pressure (Pa). Please make the statement "a high resolution mode of # RMS" clearer. What is the quantity that has the value given?

ll. 165-166: The number of GPIO pins and the WiFi and Bluetooth connectivity seem irrelevant to your application. I suggest to replace it with a description of how the HPP sensors were connected to the RPi3.

l. 169: Please clarify what you mean by "external speed adaptor".

l. 170: I suggest to replace "A 12 V power supply is sufficient to power the integrated package.", i.e. the statement of a possibility, with your actual realisation, e.g. "The package is powered by a switching power supply providing an output voltage of 12 V."

l. 183: I suggest to use $p$ and $U$ for pressure and relative humidity, respectively, to comply with the standard WMO symbols (WMO Guide to Meteorological Instruments and Methods of Observation)

l. 186: "The G2401 accuracy is estimated to be below 0.05 ppm (Rella et al., 2013)." There are two problems with this statement. Firstly, "accuracy" should be replaced by "uncertainty", because 'accuracy is a qualitative term, the numerical expression of which is uncertainty' (WMO Guide to Meteorological Instruments and Methods of Observation). Secondly, the accuracy of the G2401 analyser depends heavily on the calibration procedure, the standards used for calibration and the correction model. This information would be crucial to support the claim for 0.5 ppm uncertainty, whereas the publication of Rella et al. on water correction seems less important here.

l. 189, rows 1, column 8: suggesting "in **ambient** air" as presumably the cylinders specified in the next column are also filled with air

l. 189, rows 2 and 3, column 2: "Correlation between [CO2] and P / T" could be read as $\mathrm{corr}([\mathrm{[CO2]}, \frac{P}{T})$. The fraction $\frac{p}{T}$ is probably not intended.

ll. 203–204: Where these ambient changes or was there some kind of control? Please show the time series of both pressure and temperature during the experiment. Which was the CO2 dry air mole fraction of the air delivered to the sensors?

ll. 205–206: As far as I can see from the data sheet, the Keller 33x sensors use temperature for correction of their pressure indication, but they do not output temperature indications. Moreover, the 0.01% precision quoted is refers to full scale of the sensor. As the reader does not know which model was used in the experiments, this number alone is meaningless. Please specify the precision in Pa.

l. 216: A variation cannot be calibrated. Please reformulate this heading.

ll. 224–226: Whether dry air is the best case for the sensors depends on how they are intended to be operated in the field. Please explain this introductory sentence better or

leave it out.

Figure 2: Please redraw this diagram using standard P&ID symbols as listed e.g. here and here. In particular, the use of a 2 port valve symbol where probably a 3 port valve is meant confuses the reader. It would be worthwhile to explain the figure in the text. What is the purpose of the pump's vent? Is the CRDS analyser really connected directly to the overpressure created by the pump? *Caption:* Specify for which of the tests this setup was used – for all tests and the calibration?

l. 231: "The experimental setup is shown in Figure 2." Redundant with l. 188. After introducing the setup in the beginning of Sect. 3, I suggest to only mention the modifications, if any, in the following subsections.

ll. 261–263: Either the residuals are larger than 1 ppm or the quoted $R^2$ is too low. In any case $R^2$ is not particularly helpful in this context. Consider to calculate the mean absolute error, which can support the interpretation of measurements taken by the sensor under test.

Figure 3: Label the panels (a) to (c) (AMT figure content guidelines) and refer to these labels in the caption. Left panel: Earlier in the manuscript, "HPP3.1" refers to the first version of the series 1 instruments. Assuming that the blue line is the measurement from a single sensor as the caption suggests, please change the label to "S1.1". Upper right panel: same as left panel. Lower right panel: Plot residuals ($x_{\mathrm{HPP}} - x_{\mathrm{CRDS}}$) versus $x_{\mathrm{CRDS}}$ in addition to or in place of $x_{\mathrm{CRDS}}$ vs. $x_{\mathrm{HPP}}$. In any case, as the CRDS analyser is your reference, its measurements should be on the horizontal axis. Label the axes with the quantities displayed, not with the names of the sensors.

l. 267: "all environmental variables": There are many more environmental variables, such as irradiance or quantities related to the air surrounding the sensors. I suggest to simply name the quantities that are independent variables in the regression: pressure, temperature and humidity.

ll. 272–277: This passage can and should be simplified. In the equation, replace xP with $p$, xT with $T$, xw with $U$ and xxy with $t$ for time. Adapt the indices of the parameters $a$ and the explanation of the variables accordingly.

ll. 277-278: "Baseline [...] time." This sentence is misleading. The meaning of both $x$ and $y$ has changed compared to the two sentences before. A function $y = x$ cannot be fitted because it contains no parameters. If you rewrite the equation as suggested above and explain that $t$ is time, this sentence can be removed altogether. Otherwise, a better explanation is needed.

l. 285: I support differentiation between the "naked" sensors and complete instruments, but then it needs to be consistent. Up to here the two terms seemed to be used as synonyms. Please check all occurrences of "sensor" and "instrument".

l. 298–306: This is the third paragraph with a reference to the schematics of the experimental setup. The fact that the setup is explained here makes the reader wonder what the details of the setup were during earlier experiments? Was it the same? If yes, please move this explanation to the first paragraph in Sect. 3. If no, please state the differences precisely. It would also be helpful to mark in Fig. 2 which parts were same for all experiments and which ones were changed. Depending on the number of changes it might even be advisable to have a separate schematic for each individual setup. I am especially confused about the flow control: If the instruments were connected in parallel as shown in the schematics, how did you ensure a specific flow rate through each of them with only one flow controller?

l. 305: "each HPP was flushed every 12 hours for 30 minutes" 30 minutes per cylinder or 30 minutes for all cylinders together? Please clarify.

l. 308–310: Please split this sentence in two for clarity.

l. 358: The true value of the $CO_2$ dry air mole fraction in the cylinder is unknown, please use "assigned value"

Figure 4: *Upper panel:* Name the quantity on the vertical axes (both left and right hand side) *Lower panel:* Please explain the meaning of the black dot on the line representing the assigned value or remove it. *Caption:* "true value" -> "assigned value"

Figure 5: *Caption:* "true value" -> "assigned value"

Figure 6: *Both panels:* To convince the reader that the relationship is linear, please show a residuals plot ($x_{\mathrm{Measured}} - x_{\mathrm{LinearFit}}$ plotted vs. $p$)

Figure 7: *Both panels:* To convince the reader that the relationship is linear, please show a residuals plot ($x_{\mathrm{Measured}} - x_{\mathrm{LinearFit}}$ plotted vs. $p$)

Table 3: *Caption:* "Sensors 1 to 3 ... HPP3.2" – there is no "Sensor $n$" in the table and the relation between sensor name and generation was defined in the first paragraph of Sect. 3. To emphasise that three sensors apply their own pressure compensation you could write explicitly "S2.1, S2.2 and S2.3 have builtin pressure compensation."

ll. 404-410: The terms "standard deviation" and "standard error" (= standard deviation of the mean) seem to be used interchangeably here, although they are very different quantities. I assume that standard deviation is meant. If the distribution of the residuals is not Gaussian than the interquartile range of the residuals would be a better measure of the spread in the data. I suggest to include the calculated values in Table 2, possibly instead of $R^2$. Additional point: What is the spread after application of both the temperature and the pressure correction? Are these not the really important results? Please clarify!

l. 411: This short discussion neglects important points: (1) S1.1, S1.2 and S1.3 were tested over a much smaller pressure range than the newer sensors. (2) The second generation sensors apply the pressure correction outlined in Gaynullin et al. 2016, which is of a different form than the multivariable regression used in this article. (3) Are the standard deviations after pressure correction **significantly** different between the two sensor generations ("do not exceed 0.3 ppm" vs. "0.9, 0.2 and 0.2 ppm")?

[Figure]

ll. 424: If we write the block-wise 1-minute averages of a time series $x$ as $\overline{x}$, did you calculate the root mean square of $x_{\mathrm{HPP}} - \overline{x_{\mathrm{CRDS}}}$ or the root mean square of $\overline{x_{\mathrm{HPP}}} - \overline{x_{\mathrm{CRDS}}}$? Please clarify!

ll. 431–432: Figure 8 indicates the opposite: For a calibration interval of 6 days, the root mean square difference between the HPP measurements and the CRDS measurements is higher than 1 ppm for all sensors.

ll. 435–438: Other SenseAir sensors have a feature called Automatic Baseline Correction (ABC). Does the HPP platform use such a feature and could it be related to the observations? Additional point: Explain why two different measures (root mean square difference and mean) are used and which factors affect them. Which implications do the mean and RMS deviation have for the intended use cases of the sensors?

Figure 8: *Caption:* "an independent accurate CRDS Picarro" -> "a CRDS analyser"; "reference cylinders" -> "calibration cylinders" to be consistent with the rest of the manuscript

ll. 450–452: I suggest to leave out this sentence as it belongs to the methods. Also, "all atmospheric variables [...] which affect the performance of the instruments" might be too bold a statement (think e.g. about aerosol content).

ll. 460–466: The fact that offset correction reduces the offset to 0 is not worth noting – it is the sole purpose of this correction. The correction on the basis of CO2 dry air mole fraction is not mentioned neither in Fig. 9 nor in Table 3. If both corrections are applied in a single step, "offset correction" is a misleading name for this step. The different corrections are judged based on root mean square difference, mean difference or both, which seems arbitrary. I suggest to interpret both measures for all corrections or to use just one.

Figure 9: *Panel f):* From the text I assume that all the data shown in this figure is used for calibration. However, the curve reveals that drift is dramatically increased by

[Figure]

the drift correction. How is this possible? Why does, according to Table 3, the root mean square difference nevertheless decrease due to the drift correction? *Caption:* "the Picarro" -> "a CRDS analyser"

Table 9: *Row 2, column 2 and 3:* This looks like erroneous formatting, I assume $1.0 \cdot 10^{-3}$ and $8.5 \cdot 10^{-4}$ was meant. Why is the offset not 0 after offset correction? $10^{-3}$ is too big to blame it on floating point accuracy. *Caption:* If we write the block-wise 1-minute averages of a time series $x$ as $\overline{x}$, did you calculate the root mean square of $x_{\mathrm{HPP}} - \overline{x_{\mathrm{CRDS}}}$ or the root mean square of $\overline{x_{\mathrm{HPP}}} - \overline{x_{\mathrm{CRDS}}}$? Please clarify. This information must also be provided in the text. Additional note: Please move the sentence "A second instrument ... in the data" into the methods section.

l. 481–482: This sentence seems out of place, the connection to the preceding and following part is not clear.

l. 482: Description of a new analysis begins here, please start a new subsection. It would be helpful to state explicitly that you are again using data from a single sensor, S2.2.

ll. 484–486: Where in Fig. 10 is data for a calibration period of a single week? Additionally, "High mean $\Delta CO_2$" is not the right wording here as nearly all values in the right panel of Fig. 10 are negative.

ll. 486–487: Indicate if the results of these tests are presented somewhere.

ll. 487–489: According to Table 1, WA2-1 lasted 45 days. How could you assess the calibration with "raw measurement data not used in the learning period" when the calibration period span the entire 45 days?

ll. 497–498: "Cross-validation" is a helpful term in this context. I suggest to also use it earlier in this section.

ll. 499–502: So despite the statement in ll. 487–489 you are using the learning period for validation? This needs to be clarified! If the same data is used for training and

validation, why is the mean difference between the HPP and CRDS measurements not 0, given that your model contains a term for offset correction?

l. 501: Suddenly hourly values are used, which is confusing for the reader. Please explain somewhere between l. 483 and l. 498 the averaging that you applied. What is the rationale for using hourly values as opposed to the 1-minute averages used earlier? Please consider the experimental conditions during WA2-1 and the intended use cases of the sensors when answering this question in the text or leave out the hourly values if they are not needed. Is the learning also carried out with hourly averages?

ll. 505–510: A solid interpretation of these results is only possible if the variation of the ambient conditions is taken into account. Please provide this information in a figure, calculate interquartile ranges and revise this part accordingly.

ll. 506–507: I suggest to refer to Fig. 9, provided I am right to assume that it shows the same data.

l. 516: "and a residual slope of 0.14 and 0.28 ppm/week is shown in the black (W1) and the red (W6) curves of the figure, respectively" -> "and a residual drift of 0.14 and 0.28 ppm over one week, respectively, remains (Fig. 11)." Note that this both contains a logical improvement and avoids the mixture of "ppm" and "week" in a single unit, which is not in accordance with the rules of the SI. The latter applies to the next sentence as well.

ll. 520–523: This reads as if $\Delta CO_2$ was presented in Fig. 10, which it is not according to the legend. Whether it might change the results depends on which averages are used for the training (see comment to l. 501).

Figure 10: *Panels:* Inappropriate labels at the vertical axis (see earlier comments). The horizontal axis needs a label. The tic labels on the horizontal axis (15d, 30d, 45d, week1-week6) are unclear. Maybe the label "Learning period" and the tic labels "1st", "2nd", "3rd", "w1-6" with a clear description in the caption would be appropriate? The

legend should be made bigger for improved readability. It is as important as the axes labels, so it should be written in the same font size. *Caption:* "15 consecutive days each in x-axis" is unclear. The horizontal axes are identical for both panels, so I suggest to explain it before giving specific comments on either panel. Remove the sentence "Hourly and minute values are represented in full and empty symbols respectively.", the information is in the legend.

Figure 11: *Caption:* A term of the form $a\,(b-c)$ can be read as "a times the difference b minus c". Suggesting $\Delta x = (x_{\mathrm{CRDS}} - x_{\mathrm{HPP}})$ or directly using the difference $x_{\mathrm{CRDS}} - x_{\mathrm{HPP}}$ for the vertical axis.

ll. 541–570: Several of my comments to the previous section also apply in this section. Please check and revise.

ll. 543–548: Explanation is unclear. I suggest to first describe precisely the analysis, then the names for the different learning periods and only then referring to Fig. 12.

l. 550: "seem to provide" sounds like this observation was deceptive, but for this data set it is a fact, so I suggest replacing this phrase with just "provide".

ll. 552–554: This sentence is highly confusing because it requires reading the diagram "back in time" and involves slopes of means of differences of different training and validation periods. Please simplify, possibly leaving out the slopes.

Figure 12: *Panels:* Same issues as Fig. 10. *Caption:* Explanation unclear, "learning" and "validation" period seem to be mixed up. The horizontal axes are identical for both panels, so I suggest to explain it before giving specific comments on either panel. Remove the sentence "Hourly and minute values are represented in full and empty symbols respectively.", the information is in the legend.

ll. 576-577: "significantly decreased" and "even" contradict each other. Either stress that the sensitivity is reduced or emphasise that even the second generation sensors are sensitive to pressure.

l. 585: Remove "a process called learning", it brings machine learning to mind which is not used in this study. I suggest to generally replace all occurrences of "learning" by "calibration".

ll. 592–593: Remove "that only becomes more apparent for longer-term observations". The drift component is always in your regression model.

**3  Technical corrections**

l. 30: "month" -> "months"

l. 35: missing space in "1–2months"

l. 49: "information on emission" -> "information on emission**s**"

l. 52: missing opening parenthesis before "Mays et al."

l. 82: "in time, and in space" -> "in time and space"

l. 89: remove comma

ll. 89-91: Split sentence to make it easier understandable

l. 100: Delete "for CO2 measurements", it is redundant in this sentence. "instrument sensitivities" -> "instrument**'s** sensitivities"

l. 104: Spell out the acronym CRDS once

ll. 106–107: "Paris region environment" -> "Paris region"

l. 108: "Empirical" -> "empirical"

l. 118: missing space in "1m"

l. 121: "light in these wavelengths" -> "light at these wavelengths"

[Figure]

l. 124: Name the quantities, e.g. "an operating voltage of 12 V direct current"

l. 126: Replace comma by period

l. 128: Replace "versions" by "generations" as the term versions is later used for differentiation within the third generation

l. 140: "improve long-term drift" -> "reduce long-term drift"

l. 144: "Leakage problems impact are minimized [...]" -> "The impact of leaks on the measurement is reduced [...]"

l. 150: Remove "better"

l. 156: To measure the pressure

l. 160: The DHT22 is not made by Adafruit but by "Aosong(Guangzhou) Electronics Co., Ltd"

l. 161: "interface**s**" -> "interface"

l. 161: Provide the names of the quantities. In the data sheet of the DHT22/AM2302 the accuracy in humidity is specified as +-2%RH(Max +-5%RH). If you think that the larger uncertainty is not relevant for your application please explain why.

ll. 164–165: "The RPi3 is a small (85x56 mm) processor running with Rasbian OS which is a Linux distribution." -> "The RPi3 is a small (85 mm x 56 mm) single-board computer running Raspbian OS, a GNU/Linux distribution."

l. 185: "same air than" -> "same air **as**"

l. 186: "HPP" -> "HPP sensors". Applies at several locations in the manuscript.

l. 187: Remove "upon"

l. 188: "Figure 3" -> "Fig. **2**"

ll. 199-202: This sentence belongs to section 2.1 where the HPP sensors are described

l. 212: "temperature**s**" -> "temperature", "with linear rates of change" -> "with constant rate of change", "1 °C/hour" -> 1 °C/h

l. 214: "increment" -> "decrement"

l. 219: "operation**s**" -> "operation"

l. 228: "H2O" -> "$U$", "$0.05 \pm 0.05$ % **H2O**" -> "$0.05 \pm 0.05$ %"

l. 234: Suggesting to spell out "above ground level" as it is used only once

l. 236: "were measured successively each 13 hours during 30 minutes" is unclear. Suggestion: "Once every 13 hours, four cylinders were sampled successively for 30 min each."

l. 238: "period**s**" -> "period"

l. 267: "multivariate" -> "multivariable". A multivariate regression has more than one outcome.

l. 270: same as l. 267

l. 272: Consult AMT's guidelines on mathematical notation. Symbols should be typeset in italics (and in fact they are in the text, but not in the formula). Additional note: "$a_p x_P$": typo in the indices (small and capital P mixed).

l. 283: Remove "of the sensors"

l. 287: "outside air on top of" -> "outside air sampled on top of"

l. 294: "inner tube" -> "inner diameter"?

l. 298: "Figure 3" -> "Fig. **2**"

l. 297: "CRDS" -> "CRDS analyser" – CRDS is a technique, not a measurement device. Please check all occurrences of "CRDS" in the manuscript as to whether this replacement is applicable.

l. 301: Non-SI unit "500 mln min$^{-1}$", please correct.

l. 315: "Picarro" -> "CRDS analyser". Picarro is a company, not a measurement device. Please check all occurrences of "Picarro" in the manuscript as to whether this replacement is applicable. Remove comma before "and"

l. 317: Non-SI unit "500 mln min$^{-1}$", please correct.

l. 319: Suggesting to replace "410 ppm and minute averages varying between" with "410 ppm. 1 minute averages varied between"

l. 328: Figures 6 and 7 are referenced before Fig. 4 and 5. Please swap their positions in the manuscript.

l. 338: Use "PT2", the experiment's name, instead of "PIT", the name of the lab

Figure 4: *Caption:* "please not" -> "please not**e**"

Figure 5: *Caption:* "please not" -> "please not**e**"

l. 386: "HPP3.1 versions" -> "HPP3.1 sensors"

l. 387: "newest HPP3.2 versions" -> "newer HPP3.2 sensors"

l. 391: "that is correction" -> "that this correction"

l. 393: "range**s**" -> "range"; "-0.2 to -0.7 ppm/$^{\circ}$C" -> "-0.7 to -0.2 ppm/$^{\circ}$C"

l. 453: "Panel of Figure 9" -> "Panel a of Figure 9"

l. 505: "when learning is form the first 15 days" -> "when the first 15 days are used for validation"

l. 515: "When using the first week (W1) and the last week (W6) for learning" -> "When using either the first week (W1) or the last week (W6) for learning"

l. 542: "carried" -> "carried out"

Figure 12: *Panels:* The tic label "15d" is displaced vertically

ll. 606-608: This is a new result, please move it to the results section, possibly into a new subsection and accompanied with a few more details.

---

## Author Comment (AC1) · 12 Feb 2019

Reply: We would like to thank the reviewer for the comments and suggestions, which have guided our revisions of the manuscript

This paper reports a series of experiments evaluating one particular low cost CO2 sensor. The paper is not especially well organized. It reads as a long list of experiments. However, analysis that synthesizes the observations and context from other related work is lacking. There is an extensive knowledge base of performance for such NDIR

instruments, including manufacturer literature (e.g. LiCor, Vaisala), and field evaluation in other contexts.

Key ideas in the description of NDIR sensors that this paper could be better organized around include:

Reply: We agree that the previous version of the manuscript was indeed not organized clearly enough and we have significantly streamlined it. We have furthermore moved multiple figures into the supplemental materials.

1) that they measure absorption which is proportional to number density but the atmospheric quantity of interest is dry air mixing ratio. The measured quantity must be converted using the ideal gas law and subtracting water number density to give the dry air mixing ratio. Many of the figures are some form of confusing intermediate product along the way to a dry air mixing ratio.

Reply: This study does indeed give the different steps between the raw output of the instrument, which is supposed to be mole fractions, towards calibrated dry air mole fractions eventually. The description of the fundamental spectroscopy can be found in Hummelgard et al. 2016. Our step by step approach was deliberately chosen to highlight, which cross-sensitivities and instrument characteristics cause the deviations of this "raw" mole fraction data and calibrated data that reflects dry air mole fractions.

2) that a second order correction is associated with pressure broadening of the $CO_2$ absorption lines.

Reply: This secondary effect was not accounted for as the first order corrections allowed to achieve our repeatability target, furthermore this effect did not seem to influence the linearity of the instrument in the tested range (330-1000ppm). The new manuscript version now mentions this additional source of uncertainty.

3) that knowledge of zero is as challenging as knowledge of response to $CO_2$. The paper neglects to acknowledge or build on related work by Shusterman et al. Atmos.

Chem. Phys., 16, 13449-13463, 2016 and Zimmerman et al. Atmos. Meas. Tech., 11, 291-313 2018 and likely others.

Reply: The reason for introducing the linear drift term in the multi-variable calibration was indeed due to the issue of a non-stable zero of the instrument. We have clarified this point in the manuscript. We have also significantly extended the discussion on other work on lower-cost $CO_2$ sensors, although this paper is intended as a technical description of one specific instrument (better reflected with the new title).

Overall, I recommend a substantial revision to improve the clarity. Cutting the number of figures in half and targeting them to identified issues with performance would be welcome.

Reply: Thank you for this suggestion. The manuscript was substantially restructured to address this issue.

---

## Author Comment (AC2) · 12 Feb 2019

Reply: We would like to thank reviewer #2 for the very diligent and detailed review, which was instrumental in improving the manuscript.

General comments This manuscript reports on newly developed commercial CO2 sensors which were tested in order to assess their suitability for deployment in urban monitoring networks. The sensitivity of the low-cost sensors to environmental factors is quantified and corrected by means of a multivariable regression. The topic is of high

interest in the community and the presented work fits well the scope of AMT. The authors have carried out a series of experiments well suited to assess the accuracy that can be expected in the proposed use case.

Unfortunately, the manuscript in its current state does not realise the full potential of the extensive data set collected. Structure, precision and clarity of the language as well as formal aspects need substantial improvements. In general I strongly recommend consultation of AMT's Manuscript preparation guidelines for authors, especially the sections on manuscript composition, mathematical notation and terminology as well as the English guidelines and house standards. The structure of the manuscript suffers from loose connections between sections and a lack of coherence in the order in which information is presented. It is not easy to see which sections in the text and which figures belong together. This might be improved by referencing the short names of the corresponding experiments in every figure caption and at the beginning of each subsection.

Reply: We have substantially revised the manuscript and moved tables and figures to the supplement where suitable. Furthermore, we have revisited the language and notation concerns raised by reviewer #2.

The explanation of the different experiments in Sect. 3 needs to be extended and improved. In the current manuscript, important details are missing or the reader has to combine bits of information from different sections, tables and figures to understand how the experiment was carried out. I suggest to give all descriptions a common structure, e.g. like this: (1) Purpose of the test (2) Which sensors were tested? (3) Which air was measured, i.e. ambient or from cylinder? Was the drier used? If cylinders were used, for how long and how often was switched between them? (4) Which pumps and means of flow control were used? (5) Was there a reference measurement by a CRDS analyser? If yes, how was it connected to the system? (6) What were the ambient conditions of the sensors? In case of controlled temperature and pressure, describe the pattern. Show a graph of the ambient conditions where they matter for the experiment.

Reply: We have taken these suggestions on-board in the revised manuscript to improve readability.

All in all, make sure that the reader gets all information necessary for repeating the experiment and that this information is provided in a single section. Precision and clarity of the language is a big issue in the current form of the manuscript. Some passages are hard to understand (see Specific comments). In several places in the manuscript the value of a quantity is given without naming the quantity itself. To avoid misinterpretation, name and value of a quantity should always be provided together, e.g. "an operating voltage of 12 V direct current". See the International vocabulary of metrology – Basic and general concepts and associated terms (VIM)" (JCGM 200:2012) for details. AMT demands that "wherever possible, SI units should be used". Throughout the manuscript, "atm" should be replaced by "kPa".

Reply: We have revised the main text and where practical have used units as suggested by BIPM or the official GAW/WMO recommendations (GAW report No. 242; https://library.wmo.int/index.php?lvl=author_see&id=12735#.XGIcIWZ7mUk).

Other problematic units are listed under Specific comments and Technical corrections. I strongly urge the authors to differentiate between the substance $CO_2$ and its abundance. "$CO_2$" cannot be the quantity plotted on the axis of a diagram. What is really meant is a measure for the abundance of $CO_2$. The abundance can be expressed e.g. as a mole fraction (measured in ppm) or a concentration (measured in different units like g/L, mol/L), see e.g. the IUPAC Gold book. The expression "[A]" is commonly used for the concentration of substance A. The Picarro analysers report mole fraction, not concentration, so "[$CO_2$] from CRDS" is misleading. Mixing ratio is yet another quantity (see WMO Guide to Meteorological Instruments and Methods of Observation). As far as I can see, there is no need to refer to concentrations or mixing ratios in this manuscript and all occurrences of these quantities can be adapted such that $CO_2$ is quantified by its dry air mole fraction. Even when wet air is measured, the correction for the influence of water vapour leads to an estimation of dry air mole fraction.

**AMTD**

Reply: We have consistently changed the wording and refer to mole fractions and dry air mole fractions where appropriate.

One important factor that makes the text hard to follow is the lack of consistency in names and categories. To give an example of the problems with the current naming scheme: Sect. 3.1 is named "Laboratory tests", Sect. 3.3 is called "Field tests with urban air measurements", suggesting that laboratory and field tests are two different things. However, Table 1 has the caption "Summary of all laboratory tests", yet it contains also the measurements WA2-1 and WA2-2 which seem to be the experiments described in Sect. 3.3. Moreover, the location for WA2-1 and WA2-2 (field tests?) according to Table 2 is "Laboratory (Saclay)", while the location for PT1 (a laboratory test) is "plastic chamber (Saclay)".

Reply: We have clarified the nomenclature for the different tests and locations involved.

Please decide for either of the terms "HPP sensors" or "HPP3 sensors" when referring to all sensors that were tested. Mixing the terms confuses the reader. Likewise, please refer to specific sensors consistently with their short name defined in Sect. 2.1, e.g. "S2.2". Do not occasionally use "HPP3.2 S2.2" or similar constructs. Calibration vs. correction: It would help the clarity if the act of determining correction coefficients was consistently referred to as "calibration" and the act of applying these correction coefficients to raw measurements was consistently referred to as "correction".

Reply: We have clarified the use of those two terms. In brief "correction" refers to efforts to correct unintended instrument behavior (i.e. temperature or pressure cross-sensitivity), while calibration refers to efforts to determinate the instrument response function that allows translating the corrected raw data onto the official WMO scale for $CO_2$ dry air mole fraction.

The abbreviation RMS for root mean square is used many times in the manuscript. As far as I can see, in all instances the root mean square difference or root mean square error is actually meant. It needs to be clear which quantities are subtracted to obtain

this difference. To this end, I suggest a notation like " RMS (xS1.1 −xCRDS) = 0.5ppm ". Writing "RMS= 0.5ppm" is not acceptable. As for the formal aspects I refer the authors to the comments below and AMT's guidelines for authors.

Reply: Yes, we indeed should have referred to RMS differences, we have corrected this oversight.

Note also that authors are requested to include a statement on how the data used in the work can be accessed by others (see AMT Data policy).

Rply: We have included a statement on the data policy as well as a section describing the authors' individual contributions to the presented study.

2 Specific comments

ll. 1–2: "using commercial NDIR sensors" should be left out or moved to a different position in the title. Suggestion: "Characterization of lower-cost commercial sensors for atmospheric carbon dioxide monitoring systems in urban areas"

Reply: We have shortened and changed the title to highlight that we are only investigating one NDIR sensor type here, but have kept the word "commercial" as this is an important distinction to user studies using research and prototype sensors.

ll. 36–39: sentence should be split. Furthermore, the second clause "[...] use networks [...] for [...] networks" is not very specific and should be reformulated. Suggestion: Replace "urban CO2 networks" with the higher goal, i.e. "monitoring of urban CO2 emissions".

Reply: We have split the sentence and expanded on the intended use.

l. 22: "of a HPP commercial NDIR sensors manufactured by Senseair AB" -> "of a newly developed sensors".

Reply: Corrected

ll. 65–66: The meaning of the sentence is not clear, "only" seems to be in the wrong place. The cited reference does not include a case with four stations, please correct.

Reply: Indeed – the reference should have been Staufer et al. 2016 that performed a modelling (and inversion) study estimating CO2 fluxes for Paris. Although there are four stations monitoring only three were used when calculating urban CO2 fluxes.

l. 99: "measure CO2 based on controlling parameters for ambient air" -> "measure the dry air mole fraction of CO2 in ambient air"

Reply:We have clarified that the HPP measured mole fractions and that we use additional measurements (p, T, RH) to calculate dry air mole fractions.

ll. 102-103: A reference instrument can be calibrated over a range, but a standard has just one value. Suggestion: "[...] a suite of gas standards with CO2 dry air mole fractions between 330 and 1000 ppm [...]"

Reply:Corrected

ll. 137-139: What is the technical improvement of the sample cell redesign with respect to the topic of this publication, i.e. the use as a CO2 sensor? If there is none, please remove this point.

Reply: Removed – this information was given for completeness, but is surely not necessary for this study.

ll. 155–162: How where the in- and outlet of the pump, the HPP sensor, the pressure and temperature sensor and the air feed-through of the enclosure connected to each other? What is the response time of the sensors at the flow speeds used?

Reply: We have updated this section with the missing information.

ll. 156–157: This sentence is redundant with ll. 142–143. Remove it.

Reply: Removed – and we have added the information on the LPS performance specifications in

L142-143. ll. 157–159: Use SI units for pressure (Pa). Please make the statement "a high resolution mode of # RMS" clearer. What is the quantity that has the value given?

Reply: Change all units to hPa to comply with SI (and keep it easily readable as 1hPa is equal to 1mbar).

ll. 165-166: The number of GPIO pins and the WiFi and Bluetooth connectivity seem irrelevant to your application. I suggest to replace it with a description of how the HPP sensors were connected to the RPi3.

Reply: Moved - this information was given to highlight that other sensors can be added in the future and Wifi exist as data transfer options. We have moved this information towards the end of the manuscript as potential future additions.

l. 169: Please clarify what you mean by "external speed adaptor".

Reply: This was basically a potentiometer that allows regulating the flowrate of the pump. We removed the sentence here as this is not crucial information.

l. 170: I suggest to replace "A 12 V power supply is sufficient to power the integrated package.", i.e. the statement of a possibility, with your actual realisation, e.g. "The package is powered by a switching power supply providing an output voltage of 12 V."

Reply: Changed

l. 183: I suggest to use p and U for pressure and relative humidity, respectively, to comply with the standard WMO symbols (WMO Guide to Meteorological Instruments and Methods of Observation)

Reply: We have corrected the symbol for pressure to p, but decided to keep the more widely used RH for relative humidity as it is also used by National Metrological Institutes e.g. NIST (USA) https://www.ncbi.nlm.nih.gov/pmc/articles/PMC4751591/ as capital PSI is not familiar to everyone.

l. 186: "The G2401 accuracy is estimated to be below 0.05 ppm (Rella et al., 2013)." There are two problems with this statement. Firstly, "accuracy" should be replaced by "uncertainty", because 'accuracy is a qualitative term, the numerical expression of which is uncertainty' (WMO Guide to Meteorological Instruments and Methods of Observation). Secondly, the accuracy of the G2401 analyser depends heavily on the calibration procedure, the standards used for calibration and the correction model. This information would be crucial to support the claim for 0.5 ppm uncertainty, whereas the publication of Rella et al. on water correction seems less important here.

Reply: Thanks for catching this. We agree with the reviewer that accuracy is indeed the incorrect term here. Our calibration routine of using working gases and a set of target gases allows us to assess the short-term and long-term repeatability of the instrument. We have added the appropriate reference for these tests performed at LSCE. (https://www.atmos-meas-tech.net/8/3867/2015/amt-8-3867-2015.html)

l. 189, rows 1, column 8: suggesting "in ambient air" as presumably the cylinders specified in the next column are also filled with air

Reply: Corrected

l. 189, rows 2 and 3, column 2: "Correlation between [CO2] and P / T" could be read as corr ([[CO2],PT). The fraction p T is probably not intended.

Reply: Corrected

ll. 203–204: Where these ambient changes or was there some kind of control? Please show the time series of both pressure and temperature during the experiment. Which was the CO2 dry air mole fraction of the air delivered to the sensors?

Reply: The pressure changes were created by connecting a membrane pump to the plastic chamber, while temperature changes were induced by an air conditioning system.

ll. 205–206: As far as I can see from the data sheet, the Keller 33x sensors use temperature for correction of their pressure indication, but they do not output temperature indications. Moreover, the 0.01% precision quoted is refers to full scale of the sensor. As the reader does not know which model was used in the experiments, this number alone is meaningless. Please specify the precision in Pa.

Reply: We have added this information. Our sensor had a specified uncertainty of ca. 0.02hPA (calibrate full-scale 2bar).

l. 216: A variation cannot be calibrated. Please reformulate this heading. Reply: Corrected.

ll. 224–226: Whether dry air is the best case for the sensors depends on how they are intended to be operated in the field. Please explain this introductory sentence better or leave it out.

Reply: We have added more explanation why those two cases are discussed in the manuscript.

Figure 2: Please redraw this diagram using standard P&ID symbols as listed e.g. here and here. In particular, the use of a 2 port valve symbol where probably a 3 port valve is meant confuses the reader. It would be worthwhile to explain the figure in the text. What is the purpose of the pump's vent? Is the CRDS analyser really connected directly to the overpressure created by the pump?

Reply: We have updated the graph. The pump vent allows reducing the inlet pressure on the dryer and Picarro inlet. This allows to have constant air flow through the HPPs and the Picarro.

Caption: Specify for which of the tests this setup was used – for all tests and the calibration? l. 231: "The experimental setup is shown in Figure 2." Redundant with l. 188. After introducing the setup in the beginning of Sect. 3, I suggest to only mention the modifications, if any, in the following subsections.

Reply: Thank you for this suggestion. We have streamlined this description here.

ll. 261–263: Either the residuals are larger than 1 ppm or the quoted R2 is too low. In any case R2 is not particularly helpful in this context. Consider to calculate the mean absolute error, which can support the interpretation of measurements taken by the sensor under test. Figure 3: Label the panels (a) to (c) (AMT figure content guidelines) and refer to these labels in the caption. Left panel: Earlier in the manuscript, "HPP3.1" refers to the first version of the series 1 instruments. Assuming that the blue line is the measurement from a single sensor as the caption suggests, please change the label to "S1.1". Upper right panel: same as left panel. Lower right panel: Plot residuals (xHPP−xCRDS) versus xCRDS in addition to or in place of xCRDS vs.xHPP.

Reply: We agree that R2 is not the most useful metric here as there are only 4 points involved, but it is reported as it is very commonly used. We have updated the labelling to be consistent with the other updates in the manuscript (S1.1 instead of HP3.1) and also added a plot of the residuals in the supplemental materials, as suggested.

In any case, as the CRDS analyser is your reference, its measurements should be on the horizontal axis. Label the axes with the quantities displayed, not with the names of the sensors.

Reply: We have opted to put the reference measurements on the y-axis as they are our 'targeted' outcome. I.e. our HPP measurements are on the x-axis and an optimal correction/calibration scheme (function f(x)) will give use the correct y value.

l. 267: "all environmental variables": There are many more environmental variables, such as irradiance or quantities related to the air surrounding the sensors. I suggest to simply name the quantities that are independent variables in the regression: pressure, temperature and humidity.

Reply: Changed as suggested

ll. 272–277: This passage can and should be simplified. In the equation, replace xP with p, xT with T, xw with U and xxy with t for time. Adapt the indices of the parameters

a and the explanation of the variables accordingly.

Reply: Changed as suggested.

ll. 277-278: "Baseline [...] time." This sentence is misleading. The meaning of both x and y has changed compared to the two sentences before. A function y = x cannot be fitted because it contains no parameters. If you rewrite the equation as suggested above and explain that t is time, this sentence can be removed altogether. Otherwise, a better explanation is needed.

Reply: Removed

l. 285: I support differentiation between the "naked" sensors and complete instruments, but then it needs to be consistent. Up to here the two terms seemed to be used as synonyms. Please check all occurrences of "sensor" and "instrument".

Reply: We have changed the wording throughout the manuscript to clearly distinguish the HPP sensor and the complete instrument, i.e. HPP sensor plus additional sensors.

l. 298–306: This is the third paragraph with a reference to the schematics of the experimental setup. The fact that the setup is explained here makes the reader wonder what the details of the setup were during earlier experiments? Was it the same? If yes, please move this explanation to the first paragraph in Sect. 3. If no, please state the differences precisely. It would also be helpful to mark in Fig. 2 which parts were same for all experiments and which ones were changed. Depending on the number of changes it might even be advisable to have a separate schematic for each individual setup. I am especially confused about the flow control: If the instruments were connected in parallel as shown in the schematics, how did you ensure a specific flow rate through each of them with only one flow controller?

Reply: We have added additional schematics for all tests in the supplemental materials. The flow through all instruments was measured before the start of the experiments.

l. 305: "each HPP was flushed every 12 hours for 30 minutes" 30 minutes per cylinder

or 30 minutes for all cylinders together? Please clarify.

Reply: The HPP sensors were flushed for 30 minutes per cylinder. We have clarified this in the main text.

l. 308–310: Please split this sentence in two for clarity.

Reply: Split as suggested – (also corrected to dry air mole fraction).

l. 358: The true value of the $CO_2$ dry air mole fraction in the cylinder is unknown, please use "assigned value"

Reply: These are indeed 'assigned values'. However, the dry air mole fractions of our cylinders are traceable to the official WMO X2007 scale for $CO_2$ maintained by NOAA. This is now clarified in the manuscript

Figure 4: Upper panel: Name the quantity on the vertical axes (both left and right hand side) Lower panel: Please explain the meaning of the black dot on the line representing the assigned value or remove it.

Reply: Changed

Caption: "true value" -> "assigned value"

Reply: Change as suggested

Figure 5: Caption: "true value" -> "assigned value"

Reply: Change as suggested

Figure 6: Both panels: To convince the reader that the relationship is linear, please show a residuals plot (xMeasured−xLinearFit plotted vs. p)

Reply: Added in the supplemental materials

Figure 7: Both panels: To convince the reader that the relationship is linear, please show a residuals plot (xMeasured−xLinearFit plotted vs. p)

Reply: Added in the supplemental materials

Table 3: Caption: "Sensors 1 to 3 ... HPP3.2" – there is no "Sensor n" in the table and the relation between sensor name and generation was defined in the first paragraph of Sect. 3. To emphasise that three sensors apply their own pressure compensation you could write explicitly "S2.1, S2.2 and S2.3 have builtin pressure compensation."

Reply: We have updated Table 3 and the caption

ll. 404-410: The terms "standard deviation" and "standard error" (= standard deviation of the mean) seem to be used interchangeably here, although they are very different quantities. I assume that standard deviation is meant. If the distribution of the residuals is not Gaussian than the interquartile range of the residuals would be a better measure of the spread in the data. I suggest to include the calculated values in Table 2, possibly instead of R2. Additional point: What is the spread after application of both the temperature and the pressure correction? Are these not the really important results? Please clarify!

Reply: We had calculated IQR but we agree that this information would be worthwhile to add here specifically. We did not add the information on residuals here as this information is given in the later sections of the manuscript. We have added a reference to that section to clarify.

l. 411: This short discussion neglects important points: (1) S1.1, S1.2 and S1.3 were tested over a much smaller pressure range than the newer sensors. (2) The second generation sensors apply the pressure correction outlined in Gaynullin et al. 2016, which is of a different form than the multivariable regression used in this article. (3) Are the standard deviations after pressure correction significantly different between the two sensor generations ("do not exceed 0.3 ppm" vs. "0.9, 0.2 and 0.2 ppm")?

Reply: We have extended the discussion here to include the suggested three points.

ll. 424: If we write the block-wise 1-minute averages of a time series x as x, did

you calculate the root mean square of xHPP−xCRDS or the root mean square of xHPP−xCRDS? Please clarify!

Reply: The data was first averaged to one minute averages and then the RMS differences of those averages was calculated.

ll. 431–432: Figure 8 indicates the opposite: For a calibration interval of 6 days, the root mean square difference between the HPP measurements and the CRDS measurements is higher than 1 ppm for all sensors.

Reply: This was indeed incorrectly phrased. The 1ppm target was not achieved for 6d calibration periods or higher.

ll. 435–438: Other SenseAir sensors have a feature called Automatic Baseline Correction (ABC). Does the HPP platform use such a feature and could it be related to the observations? Additional point: Explain why two different measures (root mean square difference and mean) are used and which factors affect them. Which implications do the mean and RMS deviation have for the intended use cases of the sensors?

Reply: We do not use the ABC in this study to perform a post-correction, but have added the 'drift' of the instrument in our multi-variable calibration equation. The reason RMS and mean deviation are specifically mentioned is that have very different implications for future use of this data in inverse modelling studies. The main concern for such studies e.g. Wu et al. 2016 is that the instruments to not have a significant systematic (long-term) bias, while a larger RMSE between two instruments can be accepted. We have added a more detailed explanation on this in the introduction and refer to it here now.

Figure 8: Caption: "an independent accurate CRDS Picarro" -> "a CRDS analyser"; "reference cylinders" -> "calibration cylinders" to be consistent with the rest of the manuscript

Reply: Changed as suggested

ll. 450–452: I suggest to leave out this sentence as it belongs to the methods. Also, "all atmospheric variables [...] which affect the performance of the instruments" might be too bold a statement (think e.g. about aerosol content).

Reply: We have corrected this statement – it was indeed very imprecise.

ll. 460–466: The fact that offset correction reduces the offset to 0 is not worth noting – it is the sole purpose of this correction. The correction on the basis of $CO_2$ dry air mole fraction is not mentioned neither in Fig. 9 nor in Table 3. If both corrections are applied in a single step, "offset correction" is a misleading name for this step. The different corrections are judged based on root mean square difference, mean difference or both, which seems arbitrary. I suggest to interpret both measures for all corrections or to use just one.

Reply: Agreed – we have streamlined this discussion. However, we are not sure why reporting that the offset correction works is not worth noting.

Figure 9: Panel f): From the text I assume that all the data shown in this figure is used for calibration. However, the curve reveals that drift is dramatically increased by the drift correction. How is this possible? Why does, according to Table 3, the root mean square difference nevertheless decrease due to the drift correction?

Reply: This Figure was altogether misleading and unclear. We have removed it to shorten the manuscript as suggested by reviewer #1). There is now more detailed explanation in the main text. To clarify: Panel f actually shows the influence of the correction itself. i.e. what the data would look like without the correction. Arguably a counter-intuitive way of presenting this result.

Caption: "the Picarro" -> "a CRDS analyser" Reply: Changed

Table 9: Row 2, column 2 and 3: This looks like erroneous formatting, I assume $1.0×10−3$ and $8.5×10−4$ was meant. Why is the offset not 0 after offset correction? $10−3$ is too big to blame it on floating point accuracy.

Reply: We have revised this table.

Caption: If we write the block-wise 1-minute averages of a time series x as x, did you calculate the root mean square of xHPP−xCRDS or the root mean square of xHPP−xCRDS? Please clarify. This information must also be provided in the text. Additional note: Please move the sentence "A second instrument ... in the data" into the methods section.

Reply: We did calculate the RMS difference of the minute averages <xHPP>-<xCRDS>

l. 481–482: This sentence seems out of place, the connection to the preceding and following part is not clear.

Reply: Changed

l. 482: Description of a new analysis begins here, please start a new subsection. It would be helpful to state explicitly that you are again using data from a single sensor, S2.2.

Reply: Changed as suggested

ll. 484–486: Where in Fig. 10 is data for a calibration period of a single week? Additionally, "High mean $\Delta CO2$" is not the right wording here as nearly all values in the right panel of Fig. 10 are negative.

Reply: We corrected the wording to reflect that the mean difference from zero has increased.

ll. 486–487: Indicate if the results of these tests are presented somewhere.

Reply: Yes, different durations of calibration (learning) periods were tested and were given in Figure 10. We have decided to remove this Figure as a table better conveys the crucial information here.

ll. 487–489: According to Table 1, WA2-1 lasted 45 days. How could you assess

the calibration with "raw measurement data not used in the learning period" when the calibration period span the entire 45 days?

Reply: The total duration of the discussed experiments with 45 days 'learning' was 60 days (see line 318), i.e. 45 days were used for calibration and 15 days for validation

ll. 497–498: "Cross-validation" is a helpful term in this context. I suggest to also use it earlier in this section.

Reply: Agreed, we have added this in earlier sections.

ll. 499–502: So despite the statement in ll. 487–489 you are using the learning period for validation? This needs to be clarified! If the same data is used for training and validation, why is the mean difference between the HPP and CRDS measurements not 0, given that your model contains a term for offset correction?

Reply: The calibration (learning) and validation periods were always separate. However, we tested different configuration of calibration and validation periods. This is explained more clearly in the revised manuscript.

l. 501: Suddenly hourly values are used, which is confusing for the reader. Please explain somewhere between l. 483 and l. 498 the averaging that you applied. What is the rationale for using hourly values as opposed to the 1-minute averages used earlier? Please consider the experimental conditions during WA2-1 and the intended use cases of the sensors when answering this question in the text or leave out the hourly values if they are not needed. Is the learning also carried out with hourly averages?

Reply: The calibration (learning) is always performed on minute averages. However, we also calculate and present hourly averages as those are most commonly used in inversion studies. And have been tested in the theoretical framework for Paris by Wu et al. 2016. We have furthermore, change the nomenclature when referring to specific weeks to improve readability.

ll. 505–510: A solid interpretation of these results is only possible if the variation of the

ambient conditions is taken into account. Please provide this information in a figure, calculate interquartile ranges and revise this part accordingly.

Reply: We have added statistical information on ambient parameters (as available).

ll. 506–507: I suggest to refer to Fig. 9, provided I am right to assume that it shows the same data. l. 516: "and a residual slope of 0.14 and 0.28 ppm/week is shown in the black (W1) and the red (W6) curves of the figure, respectively" -> "and a residual drift of 0.14 and 0.28 ppm over one week, respectively, remains (Fig. 11)." Note that this both contains a logical improvement and avoids the mixture of "ppm" and "week" in a single unit, which is not in accordance with the rules of the SI. The latter applies to the next sentence as well.

Reply: The reviewers interpretation of Figure 9 is correct.

ll. 520–523: This reads as if $\Delta CO2$ was presented in Fig. 10, which it is not according to the legend. Whether it might change the results depends on which averages are used for the training (see comment to l. 501).

Reply: See comment at L 501

Figure 10: Panels: Inappropriate labels at the vertical axis (see earlier comments). The horizontal axis needs a label. The tic labels on the horizontal axis (15d, 30d, 45d, week1-week6) are unclear. Maybe the label "Learning period" and the tic labels "1st", "2nd", "3rd", "w1-6" with a clear description in the caption would be appropriate? The legend should be made bigger for improved readability. It is as important as the axes labels, so it should be written in the same font size. Caption: "15 consecutive days each in x-axis" is unclear. The horizontal axes are identical for both panels, so I suggest to explain it before giving specific comments on either panel. Remove the sentence "Hourly and minute values are represented in full and empty symbols respectively.", the information is in the legend.

Reply: Figure 10 and the caption have been reworked and moved to the supplemental

materials

Figure 11: Caption: A term of the form a (b−c) can be read as "a times the difference b minus c". Suggesting Δx=(xCRDS−xHPP) or directly using the difference xCRDS−xHPP for the vertical axis.

Reply: Corrected

ll. 541–570: Several of my comments to the previous section also apply in this section. Please check and revise.

Reply: Revised to be consistent with section 4.2.2

ll. 543–548: Explanation is unclear. I suggest to first describe precisely the analysis, then the names for the different learning periods and only then referring to Fig. 12.

Reply: Agreed and done as suggested

l. 550: "seem to provide" sounds like this observation was deceptive, but for this data set it is a fact, so I suggest replacing this phrase with just "provide".

Reply: Corrected

ll. 552–554: This sentence is highly confusing because it requires reading the diagram "back in time" and involves slopes of means of differences of different training and validation periods. Please simplify, possibly leaving out the slopes.

Reply: Agreed, we have revised and simplified this paragraph

Figure 12: Panels: Same issues as Fig. 10.

Caption: Explanation unclear, "learning" and "validation" period seem to be mixed up. The horizontal axes are identical for both panels, so I suggest to explain it before giving specific comments on either panel. Remove the sentence "Hourly and minute values are represented in full and empty symbols respectively.", the information is in the legend.

Reply: Revised

ll. 576-577: "significantly decreased" and "even" contradict each other. Either stress that the sensitivity is reduced or emphasise that even the second generation sensors are sensitive to pressure.

Reply: Changed to: "significantly decrease, however, a non-negligible pressure sensitivity remained"

l. 585: Remove "a process called learning", it brings machine learning to mind which is not used in this study. I suggest to generally replace all occurrences of "learning" by "calibration".

Reply: Changed as suggested

ll. 592–593: Remove "that only becomes more apparent for longer-term observations". The drift component is always in your regression model.

Reply: Removed

3 Technical corrections l. 30: "month" -> "months"

Reply: Corrected

l. 35: missing space in "1–2months"

Reply: Corrected

l. 49: "information on emission" -> "information on emissions"

Reply: Corrected

l. 52: missing opening parenthesis before "Mays et al." Corrected

l. 82: "in time, and in space" -> "in time and space"

Reply: Corrected

l. 89: remove comma

Reply: Corrected

ll. 89-91: Split sentence to make it easier understandable

Reply: We have updated the manuscript and also added further references.

l. 100: Delete "for CO2 measurements", it is redundant in this sentence. "instrument sensitivities" -> "instrument 's sensitivities"

Reply: Deleted and corrected

l. 104: Spell out the acronym CRDS once

Reply: Corrected

ll. 106–107: "Paris region environment" -> "Paris region"

Reply: Corrected

l. 108: "Empirical" -> "empirical"

Reply: Corrected

l. 118: missing space in "1m"

Reply: Corrected

l. 121: "light in these wavelengths" -> "light at these wavelengths"

Reply: Corrected

l. 124: Name the quantities, e.g. "an operating voltage of 12 V direct current"

Reply: Corrected

l. 126: Replace comma by period

Reply: Corrected

l. 128: Replace "versions" by "generations" as the term versions is later used for differentiation within the third generation

Reply: Corrected

l. 140: "improve long-term drift" -> "reduce long-term drift"

Reply: Corrected

l. 144: "Leakage problems impact are minimized [...]" -> "The impact of leaks on the measurement is reduced [...]"

Reply: Corrected

l. 150: Remove "better"

Reply: Removed

l. 156: To measure the pressure

Reply: The LPS pressure sensor description was moved to the section 2.1.

l. 160: The DHT22 is not made by Adafruit but by "Aosong(Guangzhou) Electronics Co., Ltd"

Reply: Our DHT22 was purchased as a kit from Adafruit and the manuals did not referred to Aosong as manufacturer. As we have no way of verifying if the sensor was produced by Aosong and if Aosong actually has the patent/licensing rights for this sensor we have changed the text to clarify that we purchased the sensor as a kit from Adafruit.

l. 161: "interfaces" -> "interface"

Reply: Corrected

l. 161: Provide the names of the quantities. In the data sheet of the DHT22/AM2302 the accuracy in humidity is specified as +-2%RH(Max +-5%RH). If you think that the

larger uncertainty is not relevant for your application please explain why.

Reply: Our specification sheet only mentioned the +/-2%rh uncertainty, we have corrected this to 2-5%rh based on further research into the specification or other DHT22 providers (re-sellers?).

ll. 164–165: "The RPi3 is a small (85x56 mm) processor running with Rasbian OS which is a Linux distribution." -> "The RPi3 is a small (85 mm x 56 mm) single-board computer running Raspbian OS, a GNU/Linux distribution."

Reply: Corrected

l. 185: "same air than" -> "same air as"

Reply: Corrected

l. 186: "HPP" -> "HPP sensors". Applies at several locations in the manuscript.

Reply: Corrected throughout the manuscript

l. 187: Remove "upon"

Reply: Corrected

l. 188: "Figure 3" -> "Fig.2"

Reply: Corrected

ll. 199-202: This sentence belongs to section 2.1 where the HPP sensors are described

Reply: Moved to section 2.1

l. 212: "temperatures" -> "temperature", "with linear rates of change" -> "with constant rate of change", "1âŮęC/hour" -> 1âŮęC/h

Reply: Corrected

l. 214: "increment" -> "decrement"

Reply: Corrected

l. 219: "operations" -> "operation"

Reply: Corrected

l. 228: "H2O" -> "U", "0.05±0.05%H2O" -> "0.05±0.05%"

Reply: Changed to RH

l. 234: Suggesting to spell out "above ground level" as it is used only once

Reply: Changed

l. 236: "were measured successively each 13 hours during 30 minutes" is unclear. Suggestion: "Once every 13 hours, four cylinders were sampled successively for 30 min each."

Reply: Changed

l. 238: "periods" -> "period"

Reply: Corrected

l. 267: "multivariate" -> "multivariable". A multivariate regression has more than one outcome.

Reply: Corrected

l. 270: same as l. 267

Reply: Corrected

l. 272: Consult AMT's guidelines on mathematical notation. Symbols should be typeset in italics (and in fact they are in the text, but not in the formula). Additional note:"apxP": typo in the indices (small and capital P mixed).

Reply: Changed as suggested

l. 283: Remove "of the sensors"

Reply: Corrected

l. 287: "outside air on top of" -> "outside air sampled on top of"

Reply: Corrected

l. 294: "inner tube" -> "inner diameter"?

Reply: Indeed, the tube to inner diameter - changed

l. 298: "Figure 3" -> "Fig.2"

Reply: Corrected

l. 297: "CRDS" -> "CRDS analyser" – CRDS is a technique, not a measurement device. Please check all occurrences of "CRDS" in the manuscript as to whether this replacement is applicable. Reply: Changed throughout the manuscript

l. 301: Non-SI unit "500 mln min-1", please correct. Reply: Change to mL min-1 Millilitre per minute is very widely used (e.g. when buying flow controllers) and 8.3x10-9 m3 per second is therefore fairly impractical information for laboratory work. In addition, mL can be used with SI (see section 6.2.8: https://physics.nist.gov/cuu/pdf/sp811.pdf) and BIPM/CIPM explicitly accepted the 'minute' for use with the International System of Units (see: SI Brochure: The International System of Units (SI) [8th edition, 2006; updated in 2014]. https://www.bipm.org/en/publications/si-brochure/table6.html

l. 315: "Picarro" -> "CRDS analyser". Picarro is a company, not a measurement device. Please check all occurrences of "Picarro" in the manuscript as to whether this replacement is applicable. Remove comma before "and"

Reply: Corrected throughout manuscript

l. 317: Non-SI unit "500 mln min-1", please correct.

Reply: See comment l.301

l. 319: Suggesting to replace "410 ppm and minute averages varying between" with "410 ppm. 1 minute averages varied between"

Reply: Changed as suggested

l. 328: Figures 6 and 7 are referenced before Fig. 4 and 5. Please swap their positions in the manuscript.

Reply: We have moved Figure X and Y into the supplemental materials to streamline the manuscript

l. 338: Use "PT2", the experiment's name, instead of "PIT", the name of the lab Figure 4: Caption: "please not" -> "please note"

Reply: Corrected

Figure 5: Caption: "please not" -> "please note"

Reply: Corrected

l. 386: "HPP3.1 versions" -> "HPP3.1 sensors"

Reply: Corrected

l. 387: "newest HPP3.2 versions" -> "newer HPP3.2 sensors"

Reply: Corrected

l. 391: "that is correction" -> "that this correction"

Reply: Corrected

l. 393: "ranges" -> "range"; "-0.2 to -0.7 ppm/âŮęC" -> "-0.7 to -0.2 ppm/âŮęC"

Reply: Corrected

l. 453: "Panel of Figure 9" -> "Panel a of Figure 9"

Reply: Changed

l. 505: "when learning is form the first 15 days" -> "when the first 15 days are used for validation"

Reply: Changed the language throughout the manuscript as "learning" is not suitable here.

l. 515: "When using the first week (W1) and the last week (W6) for learning" -> "When using either the first week (W1) or the last week (W6) for learning"

Reply: Changed the language throughout the manuscript as "learning" is not suitable here.

l. 542: "carried" -> "carried out"

Reply: Corrected

Figure 12: Panels: The tic label "15d" is displaced vertically

Reply: This was not the case in our version, we have changed the graph in the hope that this error does not reproduce.

ll. 606-608: This is a new result, please move it to the results section, possibly into a new subsection and accompanied with a few more details.

Reply: We have restructured a lot on the results and conclusion sections, including this passage.

---

## Author Response (AR2)

Comments to the Author:

Dear authors, the majority of the reviewers' comments were properly addressed. However, I stumbled upon a few minor issues which require mostly technical corrections. Some reviewers' comments were addressed in your replies but I couldn't find the respective changes in the revised version of your manuscript. My apologies in case I missed some of the respective changes in the manuscript. I realized that the track changes version did not fully match the clean, resubmitted version which made the tracking of your revisions somehow cumbersome. For example, the sentence in lines 506 to 508 is not included in the track changes version (part of doc amt-2018-329-author_response-version2.pdf) and it isn't clear in the track changes version that Figure 8 was moved to the supplementary material. Overall, the changes seem to be somehow hastily applied, also seen in the number of remaining fragments (single symbols like dots or brackets) which are still present in the revised manuscript. Thus, I urge you to once more carefully go through your manuscript to clean it up and to also check that the references to the Figures and Tables are correct. See the (most likely) incomplete list of these little incorrectnesses.

Thank you for these corrections! It seems that copy-paste errors occurred when merging the comments and corrections from different co-authors.

If not indicated otherwise, line numbers refer to the document amt-2018-329-manuscript-version4.pdf uploaded on March 11.Line 37: replace "," by "."

Lines 37 to 39: rephrase last sentence of the abstract as it uses two times "provide".

Changed

Line 109: remove "."

Changed

Line 173: remove "."
Changed

Line 181: remove "."
Changed

Reviewer #2 suggested using lower case p for pressure throughout the document. You stated that you changed it accordingly but it isn't done consistently. Please do so. See e.g. line 195, in Table 1, second column etc.

We did find a few additional uses of P instead of p, but all should be lower case now.

Line 238: add space

Added

Line 246: I suppose that you want to refer here to the WMO/GAW scale, which is correctly called "WMO $CO_2$ X2007" scale.
Corrected

Lines 299-300: "which includes all environmental variables": change to "which includes pressure, temperature and humidity", as suggested by reviewer #2 and as claimed in your reply to the reviewer that you already did so.
Changed

Line 308: remove "."
Corrected

Line 322: typo: "Jussie filed or site"
Corrected

Line 325: I recommend calling this paragraph "Saclay field site tests" as the current title rather suggests a description of the filed site and not the tests themselves.
Changed

Line 342: see comment to line 325 above.
Changed

Caption, table 2: "Slopes and intercept …" -> "Slopes and intercepts …"; change "Sensor 1 to are type HPP3.1, whereas Sensor 4 to 6 are HPP3.2." to "Sensors S1.x are type HPP3.1, whereas sensors S2.x are HPP3.2." or remove this sentence completely as it is defined above.
Changed

Line 466: "During this test (section 3.3, Saclay field site) …": please simplify: "During this test (section 3.3.1) …"
Simplified

Lines 466 - 468: See comment of reviewer #2 on lines 450-452 of the submitted version. "I suggest to leave out this sentence …" as statement is too bold. You claim in the reply that you did correct the statement but no changes were ultimately done in the revised version. Please revise this sentence. Maybe to "… {CO2} and the atmospheric variables pressure, temperature and humidity … which affect the performance … were measured …".
Changed

Line 489: add ".", remove "(".
Changed

Lines 490 to 495: I cannot find the "statistical information on ambient parameters" mentioned in your reply to reviewer #2 in response to her/his comment on line 505-510 of the submitted version.
The proper information was indeed missing in our revised manuscript. We have actually added a graph with the ambient p, T and RH data the reviewer has asked for in a new Figure (S4) including IQRs.

Line 499: reference is made to Figure 8. There is no Figure 8 in the revised manuscript.
Corrected – this refers to a Figure (S5) that is now in the supplemental materials

Line 503: reference is made to Figure 10. There is no Figure 10 in the revised manuscript. Must be Figure 7. Please doublecheck all references to the Figures.
The Figure labelling have been checked and reference to Figure 10 has been removed

Line 537: write "Jussieu ambient air measurements (WA2-2); if it is of importance that these are urban measurements, please clarify the type of the stations in section 3.3.
Changed

Table 3: please revise the formatting as suggested by reviewer #2.
Changed formatting

Regards,

Martin Steinbacher